# Preference-Based Batch and Sequential Teaching: Towards a Unified View of Models

**Farnam Mansouri**[†]   **Yuxin Chen**[‡]   **Ara Vartanian**[⋆]   **Xiaojin Zhu**[⋆]   **Adish Singla**[†]

[†]Max Planck Institute for Software Systems (MPI-SWS), {mfarnam, adishs}@mpi-sws.org,
[‡]University of Chicago, chenyuxin@uchicago.edu,
[⋆]University of Wisconsin-Madison, {aravart, jerryzhu}@cs.wisc.edu

## Abstract

Algorithmic machine teaching studies the interaction between a teacher and a learner where the teacher selects labeled examples aiming at teaching a target hypothesis. In a quest to lower teaching complexity and to achieve more natural teacher-learner interactions, several teaching models and complexity measures have been proposed for both the batch settings (e.g., worst-case, recursive, preference-based, and non-clashing models) as well as the sequential settings (e.g., local preference-based model). To better understand the connections between these different batch and sequential models, we develop a novel framework which captures the teaching process via preference functions $\Sigma$. In our framework, each function $\sigma \in \Sigma$ induces a teacher-learner pair with teaching complexity as $\mathsf{TD}(\sigma)$. We show that the above-mentioned teaching models are equivalent to specific types/families of preference functions in our framework. This equivalence, in turn, allows us to study the differences between two important teaching models, namely $\sigma$ functions inducing the strongest batch (i.e., non-clashing) model and $\sigma$ functions inducing a weak sequential (i.e., local preference-based) model. Finally, we identify preference functions inducing a novel family of sequential models with teaching complexity linear in the VC dimension of the hypothesis class: this is in contrast to the best known complexity result for the batch models which is quadratic in the VC dimension.

## 1 Introduction

Algorithmic machine teaching studies the interaction between a teacher and a learner where the teacher's goal is to find an optimal training sequence to steer the learner towards a target hypothesis [GK95, ZLHZ11, Zhu13, SBB⁺14, Zhu15, ZSZR18]. An important quantity of interest is the teaching dimension (TD) of the hypothesis class, representing the worst-case number of examples needed to teach any hypothesis in a given class. Given that the teaching complexity depends on what assumptions are made about teacher-learner interactions, different teaching models lead to different notions of teaching dimension. In the past two decades, several such teaching models have been proposed, primarily driven by the motivation to lower teaching complexity and to find models for which the teaching complexity has better connections with learning complexity measured by Vapnik–Chervonenkis dimension (VCD) [VC71] of the class.

Most of the well-studied teaching models are for the batch setting (e.g., worst-case [GK95, Kuh99], recursive [ZLHZ08, ZLHZ11, DFSZ14], preference-based [GRSZ17], and non-clashing [KSZ19] models). In these batch models, the teacher first provides a set of examples to the learner and then the learner outputs a hypothesis. In a quest to achieve more natural teacher-learner interactions and enable richer applications, various different models have been proposed for the sequential setting (e.g., local preference-based model for version space learners [CSMA⁺18], models for gradient learners [LDH⁺17, LDL⁺18, KDCS19], models inspired by control theory [Zhu18, LZZ19], models

for sequential tasks [CL12, HTS18, TGH$^+$19], and models for human-centered applications that require adaptivity [SBB$^+$13, HCMA$^+$19]).

In this paper, we seek to gain a deeper understanding of how different teaching models relate to each other. To this end, we develop a novel teaching framework which captures the teaching process via preference functions $\Sigma$. Here, a preference function $\sigma \in \Sigma$ models how a learner navigates in the version space as it receives teaching examples (see §2 for formal definition); in turn, each function $\sigma$ induces a teacher-learner pair with teaching dimension $\mathsf{TD}(\sigma)$ (see §3). We highlight some of the key results below:

- We show that the well-studied teaching models in batch setting corresponds to specific families of $\sigma$ functions in our framework (see §4 and Table 1).
- We study the differences in the family of $\sigma$ functions inducing the strongest batch model [KSZ19] and functions inducing a weak sequential model [CSMA$^+$18] (§5.2) (also, see the relationship between $\Sigma_{\mathsf{gvs}}$ and $\Sigma_{\mathsf{local}}$ in Figure 1).
- We identify preference functions inducing a novel family of sequential models with teaching complexity linear in the VCD of the hypothesis class. We provide a constructive procedure to find such $\sigma$ functions with low teaching complexity (§5.3).

Our key findings are highlighted in Figure 1 and Table 1. Here, Figure 1 illustrates the relationship between different families of preference functions that we introduce, and Table 1 summarizes the key complexity results we obtain for different families. Our unified view of the existing teaching models in turn opens up several intriguing new directions such as (i) using our constructive procedures to design preference functions for addressing open questions of whether RTD/ NCTD is linear in VCD, and (ii) understanding the notion of collusion-free teaching in sequential models. We discuss these directions further in §6.

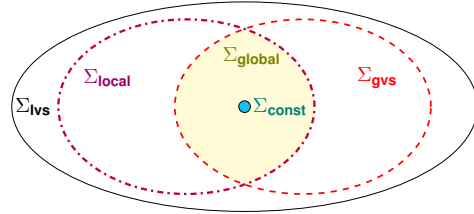

Figure 1: Venn diagram for different families of preference functions.

| Families | $\Sigma_{\mathsf{const}}$ | $\Sigma_{\mathsf{global}}$ | $\Sigma_{\mathsf{gvs}}$ | $\Sigma_{\mathsf{local}}$ | $\Sigma_{\mathsf{lvs}}$ |
|---|---|---|---|---|---|
| Reduction | TD | RTD / PBTD | NCTD | Local-PBTD | – |
| Complexity Results | – | $O(\mathsf{VCD}^2)$ | $O(\mathsf{VCD}^2)$ | $O(\mathsf{VCD}^2)$ | $O(\mathsf{VCD})$ |
| | [GK95] | [ZLHZ11, GRSZ17, HWLW17] | [KSZ19] | [CSMA$^+$18] | |

Table 1: Overview of our main results – reduction to existing models and teaching complexity.

## 2 The Teaching Model

**The teaching domain.** Let $\mathcal{X}, \mathcal{Y}$ be a ground set of unlabeled instances and the set of labels. Let $\mathcal{H}$ be a finite class of hypotheses; each element $h \in \mathcal{H}$ is a function $h : \mathcal{X} \to \mathcal{Y}$. Here, we only consider boolean functions and hence $\mathcal{Y} = \{0, 1\}$. In our model, $\mathcal{X}, \mathcal{H}$, and $\mathcal{Y}$ are known to both the teacher and the learner. There is a target hypothesis $h^\star \in \mathcal{H}$ that is known to the teacher, but not the learner. Let $\mathcal{Z} \subseteq \mathcal{X} \times \mathcal{Y}$ be the ground set of labeled examples. Each element $z = (x_z, y_z) \in \mathcal{Z}$ represents a labeled example where the label is given by the target hypothesis $h^\star$, i.e., $y_z = h^\star(x_z)$. For any $Z \subseteq \mathcal{Z}$, the *version space* induced by $Z$ is the subset of hypotheses $\mathcal{H}(Z) \subseteq \mathcal{H}$ that are consistent with the labels of all the examples, i.e., $\mathcal{H}(Z) := \{h \in \mathcal{H} \mid \forall z = (x_z, y_z) \in Z, h(x_z) = y_z\}$.

**Learner's preference function.** We consider a generic model of the learner that captures our assumptions about how the learner adapts her hypothesis based on the labeled examples received from the teacher. A key ingredient of this model is the learner's *preference function* over the hypotheses. The learner, based on the information encoded in the inputs of preference function—which include the current hypothesis and the current version space—will choose one hypothesis in $\mathcal{H}$. Our model of the learner strictly generalizes the local preference-based model considered in [CSMA$^+$18], where the learner's preference was only encoded by her current hypothesis. Formally, we consider preference functions of the form $\sigma : \mathcal{H} \times 2^{\mathcal{H}} \times \mathcal{H} \to \mathbb{R}$. For any two hypotheses $h', h''$, we say that the learner prefers $h'$ to $h''$ based on the current hypothesis $h$ and version space $H \subseteq \mathcal{H}$, iff $\sigma(h'; H, h) < \sigma(h''; H, h)$. If $\sigma(h'; H, h) = \sigma(h''; H, h)$, then the learner could pick either one of these two.

**Interaction protocol and teaching objective.** The teacher's goal is to steer the learner towards the target hypothesis $h^\star$ by providing a sequence of labeled examples. The learner starts with an initial hypothesis $h_0 \in \mathcal{H}$ before receiving any labeled examples from the teacher. At time step $t$, the teacher selects a labeled example $z_t \in \mathcal{Z}$, and the learner makes a transition from the current hypothesis to the next hypothesis. Let us denote the labeled examples received by the learner up to (and including) time step $t$ via $Z_t$. Further, we denote the learner's version space at time step $t$ as $H_t = \mathcal{H}(Z_t)$, and the learner's hypothesis before receiving $z_t$ as $h_{t-1}$. The learner picks the next hypothesis based on the current hypothesis $h_{t-1}$, version space $H_t$, and preference function $\sigma$:

$$h_t \in \underset{h' \in H_t}{\arg\min}\, \sigma(h'; H_t, h_{t-1}). \tag{2.1}$$

Upon updating the hypothesis $h_t$, the learner sends $h_t$ as feedback to the teacher. Teaching finishes here if the learner's updated hypothesis $h_t$ equals $h^\star$. We summarize the interaction in Protocol 1.[1]

---

**Protocol 1** Interaction protocol between the teacher and the learner

---

1: learner's initial version space is $H_0 = \mathcal{H}$ and learner starts from an initial hypothesis $h_0 \in \mathcal{H}$
2: **for** $t = 1, 2, 3, \ldots$ **do**
3:     learner receives $z_t = (x_t, y_t)$; updates $H_t = H_{t-1} \cap \mathcal{H}(\{z_t\})$; picks $h_t$ per Eq. (2.1);
4:     teacher receives $h_t$ as feedback from the learner;
5:     **if** $h_t = h^\star$ **then** teaching process terminates

---

## 3 The Complexity of Teaching

### 3.1 Teaching Dimension for a Fixed Preference Function

Our objective is to design teaching algorithms that can steer the learner towards the target hypothesis in a minimal number of time steps. We study the *worst-case* number of steps needed, as is common when measuring information complexity of teaching [GK95, ZLHZ11, GRSZ17, Zhu18]. Fix the ground set of instances $\mathcal{X}$ and the learner's preference $\sigma$. For any version space $H \subseteq \mathcal{H}$, the worst-case optimal cost for steering the learner from $h$ to $h^\star$ is characterized by

$$D_\sigma(H, h, h^\star) = \begin{cases} 1, & \exists z, \text{ s.t. } \mathbf{C}_\sigma(H, h, z) = \{h^*\} \\ 1 + \min_z \max_{h'' \in \mathbf{C}_\sigma(H,h,z)} D_\sigma(H \cap \mathcal{H}(\{z\}), h'', h^\star), & \text{otherwise} \end{cases}$$

where $\mathbf{C}_\sigma(H, h, z) = \arg\min_{h' \in H \cap \mathcal{H}(\{z\})} \sigma(h'; H \cap \mathcal{H}(\{z\}), h)$ denotes the set of candidate hypotheses most preferred by the learner. Note that our definition of teaching dimension is similar in spirit to the local preference-based teaching complexity defined by [CSMA+18]. We shall see in the next section, this complexity measure in fact reduces to existing notions of teaching complexity for specific families of preference functions.

Given a preference function $\sigma$ and the learner's initial hypothesis $h_0$, the teaching dimension w.r.t. $\sigma$ is defined as the worst-case optimal cost for teaching any target $h^\star$:

$$\mathsf{TD}_{\mathcal{X},\mathcal{H},h_0}(\sigma) = \max_{h^\star} D_\sigma(\mathcal{H}, h_0, h^\star). \tag{3.1}$$

### 3.2 Teaching Dimension for a Family of Preference Functions

In this paper, we will investigate several families of preference functions (as illustrated in Figure 1). For a family of preference functions $\Sigma$, we define the teaching dimension w.r.t the family $\Sigma$ as the teaching dimension w.r.t. the *best* $\sigma$ in that family:

$$\Sigma\text{-}\mathsf{TD}_{\mathcal{X},\mathcal{H},h_0} = \min_{\sigma \in \Sigma} \mathsf{TD}_{\mathcal{X},\mathcal{H},h_0}(\sigma). \tag{3.2}$$

### 3.3 Collusion-free Preference Functions

An important consideration when designing teaching models is to ensure that the teacher and the learner are "collusion-free", i.e., they are not allowed to collude or use some "coding-trick" to achieve arbitrarily low teaching complexity. A well-accepted notion of collusion-freeness in the batch setting is one proposed by [GM96] (also see [AK97, OS99, KSZ19]). Intuitively, it captures the idea that a learner conjecturing hypothesis $h$ will not change its mind when given additional information consistent with $h$. In comparison to batch models, the notion of collusion-free teaching in the sequential models is not well understood. We introduce a novel notion of collusion-freeness for the sequential setting, which captures the following idea: if $h$ is the only hypothesis in the most preferred set defined by $\sigma$, then the learner will always stay at $h$ as long as additional information received by the learner is consistent with $h$. We formalize this notion in the definition below. Note that for $\sigma$ functions corresponding to batch models (see §4), Definition 1 reduces to the collusion-free definition of [GM96].

**Definition 1 (Collusion-free preference)** *Consider a time $t$ where the learner's current hypothesis is $h_{t-1}$ and version space is $H_t$ (see Protocol 1). Further assume that the learner's preferred hypothesis for time $t$ is uniquely given by $\arg\min_{h' \in H_t} \sigma(h'; H_t, h_{t-1}) = \{\hat{h}\}$. Let $S$ be additional examples provided by an adversary from time $t$ onwards. We call a preference function collusion-free, if for any $S$ consistent with $\hat{h}$, it holds that $\arg\min_{h' \in H_t \cap \mathcal{H}(S)} \sigma(h'; H_t \cap \mathcal{H}(S), \hat{h}) = \{\hat{h}\}$.*

In this paper, we study preference functions that are collusion-free. In particular, we use $\Sigma_{\mathsf{CF}}$ to denote the set of preference functions that induce collusion-free teaching:

$$\Sigma_{\mathsf{CF}} = \{\sigma \mid \sigma \text{ is collusion-free}\}.$$

## 4 Preference-based Batch Models

### 4.1 Families of Preference Functions

We consider three families of preference functions which do not depend on the learner's current hypothesis. The first one is the family of uniform preference functions, denoted by $\Sigma_{\mathsf{const}}$, which corresponds to constant preference functions:

$$\Sigma_{\mathsf{const}} = \{\sigma \in \Sigma_{\mathsf{CF}} \mid \exists c \in \mathbb{R}, \text{ s.t. } \forall h', H, h, \sigma(h'; H, h) = c\}$$

The second family, denoted by $\Sigma_{\mathsf{global}}$, corresponds to the preference functions that do not depend on the learner's current hypothesis and version space. In other words, the preference functions capture some *global* preference ordering of the hypotheses:

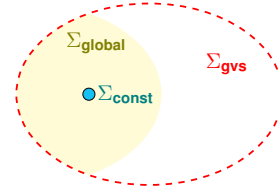

Figure 2: Batch models.

$$\Sigma_{\mathsf{global}} = \{\sigma \in \Sigma_{\mathsf{CF}} \mid \exists\, g: \mathcal{H} \to \mathbb{R}, \text{ s.t. } \forall h', H, h,\ \sigma(h'; H, h) = g(h')\}$$

The third family, denoted by $\Sigma_{\mathsf{gvs}}$, corresponds to the preference functions that depend on the learner's version space, but do not depend on the learner's current hypothesis:

$$\Sigma_{\mathsf{gvs}} = \{\sigma \in \Sigma_{\mathsf{CF}} \mid \exists\, g: \mathcal{H} \times 2^{\mathcal{H}} \to \mathbb{R}, \text{ s.t. } \forall h', H, h, \sigma(h'; H, h) = g(h', H)\}$$

Figure 2 illustrates the relationship between these preference families.

### 4.2 Complexity Results

We first provide several definitions, including the formal definition of VC dimension as well as several existing notions of teaching dimension.

**Definition 2 (Vapnik–Chervonenkis dimension [VC71])** *The VC dimension for $H \subseteq \mathcal{H}$ w.r.t. a fixed set of unlabeled instances $X \subseteq \mathcal{X}$, denoted by $\mathsf{VCD}(H, X)$, is the cardinality of the largest set of points $X' \subseteq X$ that are "shattered".[2] Formally, let $H_{|X} = \{(h(x_1), ..., h(x_n)) \mid \forall h \in H\}$ denote all possible patterns of $H$ on $X$. Then $\mathsf{VCD}(H, X) = \max |X'|$, s.t. $X' \subseteq X$ and $|H_{|X'}| = 2^{|X'|}$.*

**Definition 3 (Teaching dimension [GK95])** *For any hypothesis $h \in \mathcal{H}$, we call a set of instances $\mathrm{T}(h) \subseteq \mathcal{X}$ a teaching set for $h$, if it can uniquely identify $h \in \mathcal{H}$. The teaching dimension for $\mathcal{H}$, denoted by $TD(\mathcal{H})$, is the maximum size of the minimum teaching set for any $h \in \mathcal{H}$: $TD(\mathcal{H}) = \max_{h \in \mathcal{H}} \min |\mathrm{T}(h)|$.*

As noted by [ZLHZ08], the teaching dimension of [GK95] does not always capture the intuitive idea of cooperation between teacher and learner. The authors then introduced a model of cooperative teaching that resulted in the complexity notion of recursive teaching dimension, as defined below.

**Definition 4 (Recursive teaching dimension [ZLHZ08, ZLHZ11])** *The recursive teaching dimension (RTD) of $\mathcal{H}$, denoted by $RTD(\mathcal{H})$, is the smallest number $k$, such that one can find an ordered sequence of hypotheses in $\mathcal{H}$, denoted by $(h_1, \ldots, h_i, \ldots, h_{|\mathcal{H}|})$, where every hypothesis $h_i$ has a teaching set of size no more than $k$ to be distinguished from the hypotheses in the remaining sequence.*

In this paper we consider finite hypothesis classes. Under this setting, RTD is equivalent to preference-based teaching dimension (PBTD) [GRSZ17].

In a recent work of [KSZ19], a new notion of teaching complexity, called non-clashing teaching dimension or NCTD, was introduced (see definition below). Importantly, NCTD is the optimal teaching complexity among teaching models in the batch setting that satisfy the collusion-free property of [GM96].

**Definition 5 (Non-clashing teaching dimension [KSZ19])** *Let $\mathcal{H}$ be a hypothesis class and $\mathrm{T} : \mathcal{H} \to 2^{\mathcal{X}}$ be a "teacher mapping" on $\mathcal{H}$, i.e., mapping a given hypothesis to a teaching set.[3] We say that $\mathrm{T}$ is non-clashing on $\mathcal{H}$ iff there are no two distinct $h, h' \in \mathcal{H}$ such that $\mathrm{T}(h)$ is consistent with $h'$ and $\mathrm{T}(h')$ is consistent with $h$. The non-clashing Teaching Dimension of $\mathcal{H}$, denoted by $NCTD(\mathcal{H})$, is defined as $NCTD(\mathcal{H}) = \min_{\mathrm{T} \text{ is non-clashing}} \{\max_{h \in \mathcal{H}} |\mathrm{T}(h)|\}$.*

We show in the following, that the teaching dimension $\Sigma$-TD in Eq. (3.2) unifies the above definitions of TD's for batch models.

**Theorem 1 (Reduction to existing notions of TD's)** *Fix $\mathcal{X}, \mathcal{H}, h_0$. The teaching complexity for the three families reduces to the existing notions of teaching dimensions:*

1. $\Sigma_{const}\text{-}TD_{\mathcal{X},\mathcal{H},h_0} = TD(\mathcal{H})$
2. $\Sigma_{global}\text{-}TD_{\mathcal{X},\mathcal{H},h_0} = RTD(\mathcal{H}) = O(VCD(\mathcal{H},\mathcal{X})^2)$
3. $\Sigma_{gvs}\text{-}TD_{\mathcal{X},\mathcal{H},h_0} = NCTD(\mathcal{H}) = O(VCD(\mathcal{H},\mathcal{X})^2)$

Our teaching model strictly generalizes the local-preference based model of [CSMA$^+$18], which reduces to the "worst-case" model when $\sigma \in \Sigma_{const}$ (corresponding to TD) [GK95] and the global "preference-based" model when $\sigma \in \Sigma_{global}$. Hence we get $\Sigma_{const}\text{-}TD_{\mathcal{X},\mathcal{H},h_0} = TD(\mathcal{H})$ and $\Sigma_{global}\text{-}TD_{\mathcal{X},\mathcal{H},h_0} = RTD(\mathcal{H})$. To establish the equivalence between $\Sigma_{gvs}\text{-}TD_{\mathcal{X},\mathcal{H},h_0}$ and $NCTD(\mathcal{H})$, it suffices to show that for any $\mathcal{X}, \mathcal{H}, h_0$, the following holds: (i) $\Sigma_{gvs}\text{-}TD_{\mathcal{X},\mathcal{H},h_0} \geqslant NCTD(\mathcal{H})$, and (ii) $\Sigma_{gvs}\text{-}TD_{\mathcal{X},\mathcal{H},h_0} \leqslant NCTD(\mathcal{H})$. The full proof is provided in Appendix A.2 of the supplementary.

In Table 2, we consider the well known Warmuth hypothesis class [DFSZ14] where $\Sigma_{const}\text{-}TD = 3$, $\Sigma_{global}\text{-}TD = 3$, and $\Sigma_{gvs}\text{-}TD = 2$. Table 2b and Table 2d show preference functions $\sigma \in \Sigma_{const}$, $\sigma \in \Sigma_{global}$, and $\sigma \in \Sigma_{gvs}$ that achieve the minima in Eq. (3.2). Table 2a shows the teaching sequences achieving these teaching dimensions for these preference functions. In Appendix A.1, we provide another hypothesis class where $\Sigma_{const}\text{-}TD = 3$, $\Sigma_{global}\text{-}TD = 2$, and $\Sigma_{gvs}\text{-}TD = 1$.

## 5 Preference-based Sequential Models

### 5.1 Families of Preference Functions

In this section, we investigate two families of preference functions that depend on the learner's current hypothesis $h_{t-1}$. The first one is the family of local preference-based functions [CSMA$^+$18], denoted by $\Sigma_{local}$, which corresponds to preference functions that depend on the learner's current (local) hypothesis, but do not depend on the learner's version space:

$$\Sigma_{local} = \{\sigma \in \Sigma_{CF} \mid \exists\, g : \mathcal{H} \times \mathcal{H} \to \mathbb{R}, \text{ s.t. } \forall h', H, h, \sigma(h'; H, h) = g(h', h)\}$$

| $h$ \ $x$ | $x_1$ | $x_2$ | $x_3$ | $x_4$ | $x_5$ | $\mathcal{S}_{\mathsf{const}} = \mathcal{S}_{\mathsf{global}}$ | $\mathcal{S}_{\mathsf{gvs}}$ | $\mathcal{S}_{\mathsf{local}}$ | $\mathcal{S}_{\mathsf{lvs}}$ |
|---|---|---|---|---|---|---|---|---|---|
| $h_1$ | 1 | 1 | 0 | 0 | 0 | $(x_1, x_2, x_4)$ | $(x_1, x_2)$ | $(x_1)$ | $(x_1)$ |
| $h_2$ | 0 | 1 | 1 | 0 | 0 | $(x_2, x_3, x_5)$ | $(x_2, x_3)$ | $(x_3)$ | $(x_2)$ |
| $h_3$ | 0 | 0 | 1 | 1 | 0 | $(x_1, x_3, x_4)$ | $(x_3, x_4)$ | $(x_3, x_4)$ | $(x_3)$ |
| $h_4$ | 0 | 0 | 0 | 1 | 1 | $(x_2, x_4, x_5)$ | $(x_4, x_5)$ | $(x_5, x_4)$ | $(x_4)$ |
| $h_5$ | 1 | 0 | 0 | 0 | 1 | $(x_1, x_3, x_5)$ | $(x_1, x_5)$ | $(x_5)$ | $(x_5)$ |
| $h_6$ | 1 | 1 | 0 | 1 | 0 | $(x_1, x_2, x_4)$ | $(x_2, x_4)$ | $(x_4)$ | $(x_3)$ |
| $h_7$ | 0 | 1 | 1 | 0 | 1 | $(x_2, x_3, x_5)$ | $(x_3, x_5)$ | $(x_3, x_5)$ | $(x_4)$ |
| $h_8$ | 1 | 0 | 1 | 1 | 0 | $(x_1, x_3, x_4)$ | $(x_1, x_4)$ | $(x_4, x_3)$ | $(x_5)$ |
| $h_9$ | 0 | 1 | 0 | 1 | 1 | $(x_2, x_4, x_5)$ | $(x_2, x_5)$ | $(x_4, x_5)$ | $(x_1)$ |
| $h_{10}$ | 1 | 0 | 1 | 0 | 1 | $(x_1, x_3, x_5)$ | $(x_1, x_3)$ | $(x_5, x_3)$ | $(x_2)$ |

(a) The Warmuth hypothesis class and the corresponding teaching sequences (denoted by $\mathcal{S}$).

| $h'$ | $\forall h' \in H$ |
|---|---|
| $\sigma_{\mathsf{const}}(h'; \cdot, \cdot)$ | 0 |
| $\sigma_{\mathsf{global}}(h'; \cdot, \cdot)$ | |

(b) $\sigma_{\mathsf{const}}$ and $\sigma_{\mathsf{global}}$

| $h \backslash h'$ | $h_1$ | $h_2$ | $h_3$ | $h_4$ | $h_5$ | $h_6$ | $h_7$ | $h_8$ | $h_9$ | $h_{10}$ |
|---|---|---|---|---|---|---|---|---|---|---|
| $\sigma_{\mathsf{local}}(h'; \cdot, h = h_1)$ | 0 | 2 | 4 | 4 | 2 | 1 | 3 | 3 | 3 | 3 |
| $\cdots$ | | | | | | | | | | |

(c) $\sigma_{\mathsf{local}}$ representing the Hamming distance between $h'$ and $h$.

| $h'$ | $h_1$ | $h_2$ | $\cdots$ |
|---|---|---|---|
| $H$ | $\{h_1, h_6\}$ | $\{h_2, h_7\}$ | $\cdots$ |
| | $\{h_1\}$ | $\{h_2\}$ | $\cdots$ |
| $\sigma_{\mathsf{gvs}}$ | 0 | 0 | $\cdots$ |

(d) $\sigma_{\mathsf{gvs}}(h'; H, \cdot)$

| $h'$ | $h_1$ | $h_2$ | | $\cdots$ |
|---|---|---|---|---|
| $H$ | $\{h_1\} \cup \{h_5, h_6, h_8, h_{10}\}^*$ | $\{h_2\} \cup \{h_1, h_7, h_6, h_9\}^*$ | | $\cdots$ |
| $h$ | $h_1$ | $h_1$ | $h_2$ | $\cdots$ |
| $\sigma_{\mathsf{lvs}}$ | 0 | 0 | 0 | $\cdots$ |

(e) $\sigma_{\mathsf{lvs}}(h'; H, h)$. Here, $\{\cdot\}^*$ denotes all subsets.

Table 2: Teaching sequences with different preference functions for the Warmuth hypothesis class [DFSZ14].[4] Full preference functions are given in Appendix B of the supplementary.

The second family, denoted by $\Sigma_{\mathsf{lvs}}$, corresponds to the preference functions that depend on all three arguments of $\sigma(h'; H, h)$. The dependence of $\sigma$ on the learner's current (local) hypothesis and the version space renders a powerful family of preference functions:

$$\Sigma_{\mathsf{lvs}} = \{\sigma \in \Sigma_{\mathsf{CF}} \mid \exists\, g : \mathcal{H} \times 2^{\mathcal{H}} \times \mathcal{H} \to \mathbb{R}, \text{ s.t. } \forall h', H, h, \sigma(h'; H, h) = g(h', H, h)\}$$

Figure 1 illustrates the relationship between these preference families. As an example, in Table 2c and Table 2e, we provide the preference functions $\sigma_{\mathsf{local}}$ and $\sigma_{\mathsf{lvs}}$ for the Warmuth hypothesis class that achieve the minima in Eq. (3.2).

## 5.2 Comparing $\Sigma_{\mathsf{gvs}}$-TD and $\Sigma_{\mathsf{local}}$-TD

In the following, we show that substantial differences arise as we transition from $\sigma$ functions inducing the strongest batch (i.e., non-clashing) model to $\sigma$ functions inducing a weak sequential (i.e., local preference-based) model. We provide the full proof of Theorem 2 in Appendix C of the supplementary.

**Theorem 2** *Neither of the families $\Sigma_{gvs}$ and $\Sigma_{local}$ dominates the other. Specifically,*

1. *$\Sigma_{gvs} \cap \Sigma_{local} = \Sigma_{global}$*
2. *There exist $\mathcal{H}, \mathcal{X}$, where $\forall h_0 \in \mathcal{H}, \Sigma_{local}\text{-}TD_{\mathcal{X},\mathcal{H},h_0} > \Sigma_{gvs}\text{-}TD_{\mathcal{X},\mathcal{H},h_0}$*
3. *There exist $\mathcal{H}, \mathcal{X}$, where $\forall h_0 \in \mathcal{H}, \Sigma_{local}\text{-}TD_{\mathcal{X},\mathcal{H},h_0} < \Sigma_{gvs}\text{-}TD_{\mathcal{X},\mathcal{H},h_0}$*

## 5.3 Complexity Results

We now connect the teaching complexity of the sequential models with the VC dimension.

**Theorem 3** *$\Sigma_{local}\text{-}TD_{\mathcal{X},\mathcal{H},h_0} = O(VCD(\mathcal{H},\mathcal{X})^2)$, and $\Sigma_{lvs}\text{-}TD_{\mathcal{X},\mathcal{H},h_0} = O(VCD(\mathcal{H},\mathcal{X}))$.*

To establish the proof, we first introduce an important definition (Definition 6) and a key lemma (Lemma 4).

**Definition 6 (Compact-Distinguishable Set)** *Fix $H \subseteq \mathcal{H}$ and $X \subseteq \mathcal{X}$, where $X = \{x_1, ..., x_n\}$. Let $H_{|X} = \{(h(x_1), ..., h(x_n)) \mid \forall h \in H\}$ denote all possible patterns of $H$ on $X$. Then, we say that $X$ is* compact-distinguishable *on $H$, if $|H_{|X}| = |H|$ and $\forall X' \subset X$, $|H_{|X'}| < |H|$. We will use $\Psi_H$ to denote a compact-distinguishable set on $H$.*

In words, one can uniquely identify any hypothesis in $H$ with a (sub)set of examples from $\Psi_H$ (also see the definition of distinguishing sets in [DFSZ14]). Our definition of compact-distinguishable set further implies that there are no "redundant" examples in $\Psi_H$. It can be shown that a compact-distinguishable set satisfies the following two properties: (i) it does not contain any pair of distinct instances $x, x'$ such that $(\forall h \in H : h(x) = h(x'))$ or $(\forall h \in H : h(x) \neq h(x'))$; and (ii) it does not contain any instance $x$ such that $(\forall h \in H : h(x) = 1)$ or $(\forall h \in H : h(x) = 0)$.

**Lemma 4** *Consider a subset $H \subseteq \mathcal{H}$ and any compact-distinguishable set $\Psi_H = \{x_1, ..., x_{|\Psi_H|}\}$. Fix any hypothesis $h_H \in H$. Let $d = \mathsf{VCD}(H, \Psi_H)$ denote the VC dimension of $H$ on $\Psi_H$. If $d \geqslant 1$, we can divide $H$ into $m = |\Psi_H| + 1$ separate hypothesis classes $\{H^1, ..., H^m\}$, such that*

*(i) $\forall j \in [m]$, there exists a compact-distinguishable set $\Psi_{H^j}$ s.t. $\mathsf{VCD}(H^j, \Psi_{H^j}) \leqslant d - 1$.*

*(ii) $\forall j \in [m-1]$, $H^j$ is not empty and $H^j_{|\{x_j\}} = \{(1 - h_H(x_j))\}$.*

*(iii) $H^m = \{h_H\}$.*

Lemma 4 suggests that for any $\mathcal{H}, \mathcal{X}$, one can partition the hypothesis class $\mathcal{H}$ into $m \leqslant |\mathcal{X}| + 1$ subsets with lower VC dimension with respect to some compact-distinguishable set.[5] The main idea of the lemma is similar to the reduction of a concept class w.r.t. some instance $x$ to lower $\mathsf{VCD}$ as done in Theorem 9 of [FW95]. The key distinction of Lemma 4 is that we consider compact-distinguishable sets for this partitioning, which in turn ensures the uniqueness of the version spaces associated with these partitions (see proof of Theorem 3). Another key novelty in our proof of Theorem 3 is to recursively apply the reduction step from the lemma.

To prove the lemma, we provide a constructive procedure to partition the hypothesis class, and show that the resulting partitions have reduced VC dimensions on some compact-distinguishable set. We highlight the procedure for constructing the partitions in Algorithm 2 (Line 7– Line 10). In Figure 3, we provide an illustrative example for creating such partitions for the Warmuth hypothesis class from Table 2a. We sketch the proof of Lemma 4 below, and defer the detailed proof to Appendix D.1.

**Proof** [Proof Sketch of Lemma 4] Let us define $H_x = \{h \in H : h \triangle x_{|\Psi_H} \in H_{|\Psi_H}\}$. Here, $h \triangle x$ denotes the hypothesis that only differs with $h$ on the label of $x$, and $h_{|\Psi_H}$ denotes the patterns of $h$ on $\Psi_H$. Fix a reference hypothesis $h_H$. For all $j \in [m-1]$, let $y_j = 1 - h_H(x_j)$ be the opposite label of $x_j \in \Psi_H$ as provided by $h_H$. As shown in Line 9 of Algorithm 2, we consider the set $H^1 := H^{y_1}_{x_1} = \{h \in H_{x_1} : h(x_1) = y_1\}$ as the first partition. In the appendix, we show that $|H^1| > 0$.

Next, we show that $\mathsf{VCD}(H^1, \Psi_H \backslash \{x_1\}) \leqslant d - 1$. When $d > 1$, we prove the statement as follows:

$$\mathsf{VCD}(H^1, \Psi_H \backslash \{x_1\}) \leqslant \mathsf{VCD}(H^{y_1}_{x_1}, \Psi_H) = \mathsf{VCD}(H_{x_1}, \Psi_H) - 1 \leqslant \mathsf{VCD}(H, \Psi_H) - 1 \leqslant d - 1$$

In the appendix, we prove the statement for $d = 1$, and further show that there exists a compact-distinguishable set $\Psi_{H^1} \subseteq \Psi_H \backslash \{x_1\}$ for the first partition $H^1$. Then, we conclude that the first partition $H^1$ has $\mathsf{VCD}(H^1, \Psi_{H^1}) \leqslant d - 1$.

Next, we remove the first partition $H^1$ from $H$, and continue to create the above mentioned partitions on $H_{\text{rest}} = H \backslash H^1$ and $X_{\text{rest}} = \Psi_H \backslash \{x_1\}$. As discussed in the appendix, we show that $X_{\text{rest}}$ is a compact-distinguishable set on $H_{\text{rest}}$. Therefore, we can repeat the above procedure (Line 7– Line 10, Algorithm 2) to create the subsequent partitions. This process continues until the size of $X_{\text{rest}}$ reduces to 1, i.e. $X_{\text{rest}} = \{x_{m-1}\}$. Until then, we obtain partitions $\{H^1, ..., H^{m-2}\}$. By construction, $H^j$ satisfy properties (i) and (ii) for all $j \in [m-2]$.

It remains to show that $H^{m-1}$ and $H^m$ also satisfy the properties in Lemma 4. Since $X_{\text{rest}} = \{x_{m-1}\}$ before we start iteration $m-1$, and $X_{\text{rest}}$ is a compact-distinguishable set for $H_{\text{rest}}$, there must exist exactly two hypotheses in $H_{\text{rest}}$, and therefore $|H^{m-1}|, |H^m| = 1$. This implies that $\mathsf{VCD}(H^{m-1}, \Psi_{H^{m-1}}) = \mathsf{VCD}(H^m, \Psi_{H^m}) = 0$. Furthermore, $\forall j \in [m-1]$ and $h \in H^j$, we have $h_H(x_j) \neq h(x_j)$. This indicates $h_H \in H_m$, and hence $H_m = \{h_H\}$ which completes the proof. ∎

**Algorithm 2** Recursive procedure for constructing $\sigma_{\mathsf{lvs}}$ achieving $\mathsf{TD}_{\mathcal{X},\mathcal{H},h_0}(\sigma_{\mathsf{lvs}}) \leqslant \mathsf{VCD}(\mathcal{H},\mathcal{X})$

**Input:** $\mathcal{X}, \mathcal{H}, h_0$

1: Let $I : \mathcal{H} \to \{1, \ldots, |\mathcal{H}|\}$ be any bijective mapping
2: For all $h' \in \mathcal{H}, H \subseteq \mathcal{H}, h \in \mathcal{H}$, initialize

$$\sigma_{\mathsf{lvs}}(h'; H, h) \leftarrow \begin{cases} 0 & \text{if } h' = h \\ |\mathcal{H}| + 1 & \text{o.w.} \end{cases}$$

3: SETPREFERENCE$(\mathcal{H}, \mathcal{H}, \mathcal{X}, h_0)$
4: **function** SETPREFERENCE$(V, H, X, h)$
5:     Create compact-distinguishable set $\Psi_H \subseteq X$
6:     $H_{\mathrm{rest}} := H, X_{\mathrm{rest}} := \Psi_H$
7:     **for** $x \in \Psi_H$ **do**
8:         $y = 1 - h(x)$
9:         $H_x^y \leftarrow \{h' \in H_{\mathrm{rest}} : h' \triangle x_{|X_{\mathrm{rest}}} \in H_{\mathrm{rest}|X_{\mathrm{rest}}}, h'(x) = y\}$
10:        $H_{\mathrm{rest}} \leftarrow H_{\mathrm{rest}} \backslash H_x^y, X_{\mathrm{rest}} \leftarrow X_{\mathrm{rest}} \backslash \{x\}$
11:        $V_{\mathrm{next}} \leftarrow V \cap \mathcal{H}(\{(x, y)\})$
12:        **for** $h' \in H_x^y$ **do** $\sigma_{\mathsf{lvs}}(h'; V_{\mathrm{next}}, h) \leftarrow I(h')$
13:        $h_{\mathrm{next}} \leftarrow \arg\min_{h' \in H_x^y} I(h')$
14:        SETPREFERENCE$(V_{\mathrm{next}}, H_x^y, \Psi_H \backslash \{x\}, h_{\mathrm{next}})$

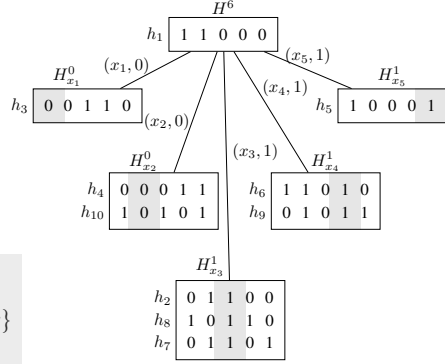

Figure 3: Illustration of Lemma 4 on the Warmuth class. The grouped hypotheses in the leaf clusters correspond to the sets $H_x^y$ created in Line 9 of Algorithm 2.

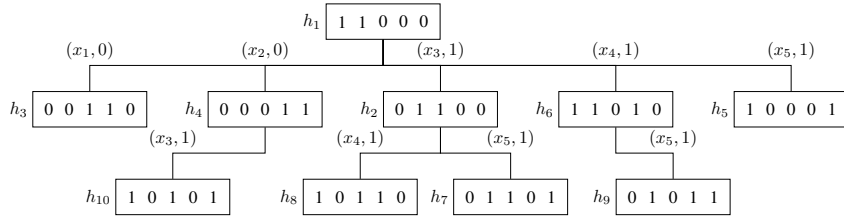

Figure 4: Illustration of Theorem 3 proof – constructing a $\sigma_{\mathsf{lvs}} \in \Sigma_{\mathsf{lvs}}$ for the Warmuth class.

**Recursive construction of $\sigma_{\mathsf{lvs}}$.** As a part of the Theorem 3 proof, we provide a recursive procedure for constructing a $\sigma_{\mathsf{lvs}} \in \Sigma_{\mathsf{lvs}}$ achieving $\mathsf{TD}_{\mathcal{X},\mathcal{H},h_0}(\sigma_{\mathsf{lvs}}) = O\left(\mathsf{VCD}(\mathcal{H}, \mathcal{X})\right)$.

**Proof** [Proof of Theorem 3] In a nutshell, the proof consists of three steps: (i) initialization of $\sigma_{\mathsf{lvs}}$, (ii) setting the preferences by recursively invoking the constructive procedure for Lemma 4, and (iii) showing that there exists a teaching sequence of length up to $d$ for any target hypothesis $h^\star$. We summarize the recursive procedure in Algorithm 2.

*Step (i).* To begin with, we initialize $\sigma_{\mathsf{lvs}}$ with default values which induce high $\sigma$ values (i.e., low preference), except for $\sigma(h'; H, h) = 0$ where $h' = h$ (c.f. Line 2 of Algorithm 2). The self-preference guarantees that $\sigma_{\mathsf{lvs}}$ is collusion-free as per Definition 1.

*Step (ii).* The recursion begins at the top level with $H = \mathcal{H}$, current version space $V = \mathcal{H}$, and initial hypothesis $h = h_0$. Lemma 4 suggests that we can partition $H$ into $m = |\Psi_H| + 1$ groups $\{H^1, ..., H^m\}$, where for all $j \in [m]$, there exists a compact-distinguishable set $\Psi_{H^j}$ that satisfies the properties in Lemma 4.

Now consider the hypothesis $h := h_0$. We show that for $j \in [m-1]$, every $(x_j, y_j)$, where $x_j \in \Psi_H$ and $y_j = 1 - h(x_j)$, corresponds to a unique version space $V^j := \{h \in V : h(x_j) = y_j\}$. To prove this statement, we consider $R^j := V^j \cap H = \{h \in H : h(x_j) = y_j\}$. As is discussed in Appendix D.2 of the supplementary, we know that none of $R^j$ for $j \in [m-1]$ are equal. This indicates that none of $V^j$ for $j \in [m-1]$ are equal.

We then set the values of the preference function $\sigma_{\mathsf{lvs}}(\cdot; V^j, h)$ for all $j \in [m-1]$ and $y_j = 1 - h(x_j)$ (Line 12). Upon receiving $(x_j, y_j)$, the learner will be steered to the next "search space" $H^j$, with version space $V^j$. By Lemma 4 we have $\mathsf{VCD}(H^j, \Psi_{H^j}) \leqslant \mathsf{VCD}(H, \Psi_H) - 1$.

We will build the preference function $\sigma_{\mathsf{lvs}}$ recursively $m - 1$ times for each $(V^j, H^j, \Psi_{H^j}, h_{\mathrm{next}})$, where $h_{\mathrm{next}}$ corresponds to the unique hypothesis identified by function $I$ (Line 13–Line 14). At

each level of recursion, $\mathsf{VCD}$ reduces by 1. We stop the recursion when $\mathsf{VCD}(H^j; \Psi_{H^j}) = 0$, which corresponds to the scenario $|H^j| = 1$.

*Step (iii).* Given the preference function constructed in Algorithm 2, we can build up the set of (labeled) teaching examples recursively. Consider the beginning of the teaching process, where the learner's current hypothesis is $h_0$ and version space is $\mathcal{H}$, and the goal of the teacher is to teach $h^\star$. Consider the first level of the recursion in Algorithm 2, where we divide $\mathcal{H}$ into $m = |\Psi_{\mathcal{H}}| + 1$ groups $\{H^1, ..., H^m\}$. Let us consider the case where $h^\star \in H^{j^\star}$ with $j^\star \in [m-1]$. The teacher provides an example given by $(x = x_{j^\star}, y = h^\star(x_{j^\star}))$. After receiving the teaching example, the resulting partition $H^{j^\star}$ will stay in the version space; meanwhile, $h_0$ will be removed from the version space. The new version space will be $V^{j^\star}$. The learner's new hypothesis induced by the preference function is given by $h_{\text{next}} \in H^{j^\star}$. By repeating this teaching process for a maximum of $d$ steps, the learner reaches a partition of size 1 (see *Step (ii)* for details). At this step $h^\star$ must be the only hypothesis left in the search space. Therefore, $h_{\text{next}} = h^\star$, and the learner has reached $h^\star$. ∎

Figure 4 illustrates the recursive construction of a $\sigma_{\text{lvs}} \in \Sigma_{\text{lvs}}$ for the Warmuth class, with $\mathsf{TD}_{\mathcal{X}, \mathcal{H}, h_0}(\sigma_{\text{lvs}}) = 2$.

## 6    Discussion and Conclusion

We now discuss a few thoughts related to different families of preference functions. First of all, the size of the families grows exponentially as we change our model from $\Sigma_{\text{const}}$, $\Sigma_{\text{global}}$ to $\Sigma_{\text{gvs}}/\Sigma_{\text{local}}$ and finally to $\Sigma_{\text{lvs}}$, thus resulting in more powerful models with lower teaching complexity. While run time has not been the focus of this paper, it would be interesting to characterize the presumably increased run time complexity of sequential learners and teachers with complex preference functions. Furthermore, as the size of the families grow, the problem of finding the best preference function $\sigma$ in a given family $\Sigma$ that achieve the minima in Eq. (3.2) becomes more computationally challenging.

The recursive procedure in Algorithm 2 creates a preference function $\sigma_{\text{lvs}} \in \Sigma_{\text{lvs}}$ that has teaching complexity at most $\mathsf{VCD}$. It is interesting to note that the resulting preference function $\sigma_{\text{lvs}}$ has the characteristic of "win-stay, loose shift" [BDGG14, CSMA$^+$18]: Given that for any hypothesis we have $\sigma(h; \cdot, h) = 0$, the learner prefers her current hypothesis as long as it remains consistent. Preference functions with this characteristic naturally exhibit the collusion-free property in Definition 1. For some problems, one can achieve lower teaching complexity for a $\sigma \in \Sigma_{\text{lvs}}$. In fact, the preference function $\sigma_{\text{lvs}}$ we provided for the Warmuth class in Table 2e has teaching complexity 1, while the preference function constructed in Figure 4 has teaching complexity 2.

One fundamental aspect of modeling teacher-learner interactions is the notion of collusion-free teaching. Collusion-freeness for the batched setting is well established in the research community and $\mathsf{NCTD}$ characterizes the complexity of the strongest collusion-free batch model. In this paper, we are introducing a new notion of collusion-freeness for the sequential setting (Definition 1). As discussed above, a stricter condition is the "win-stay lose-shift" model, which is easier to validate without running the teaching algorithm. In contrast, the condition of Definition 1 is more involved in terms of validation and is a joint property of the teacher-learner pair. One intriguing question for future work is defining notions of collusion-free teaching in sequential models and understanding their implications on teaching complexity.

Another interesting direction of future work is to better understand the properties of the teaching parameter $\Sigma$-$\mathsf{TD}$. One question of particular interest is showing that the teaching parameter is not upper bounded by any constant independent of the hypothesis class, which would suggest a strong collusion in our model. We can show that for certain hypothesis classes, $\Sigma$-$\mathsf{TD}$ is lower bounded by a function of $\mathsf{VCD}$. In particular, for the power set class of size $d$ (which has $\mathsf{VCD} = d$), $\Sigma$-$\mathsf{TD}$ is lower bounded by $\Omega\left(\frac{d}{\log d}\right)$. Another direction of future work is to understand whether this parameter is additive or subadditive over disjoint domains. Also, we consider a generalization of our results to the infinite VC classes as a very interesting direction for future work.

Our framework provides novel tools for reasoning about teaching complexity by constructing preference functions. This opens up an interesting direction of research to tackle important open problems, such as proving whether $\mathsf{NCTD}$ or $\mathsf{RTD}$ is linear in $\mathsf{VCD}$ [SZ15, CCT16, HWLW17, KSZ19]. In this paper, we showed that neither of the families $\Sigma_{\text{gvs}}$ and $\Sigma_{\text{local}}$ dominates the other (Theorem 2). As a direction for future work, it would be important to further quantify the complexity of $\Sigma_{\text{local}}$ family.

## Acknowledgements

This work was done in part when Yuxin Chen was at Caltech. Xiaojin Zhu is supported by NSF 1545481, 1561512, 1623605, 1704117, 1836978 and the MADLab AF CoE FA9550-18-1-0166.

## Footnotes

[1]It is important to note that in our teaching model, the teacher and the learner use the same preference function. This assumption of shared knowledge of the preference function is also considered in existing teaching models for both the batch settings (e.g., as in [ZLHZ11, GRSZ17]) and the sequential settings [CSMA+18].

[2]In the classical definition of $\mathsf{VCD}$, only the first argument $H$ is present; the second argument $X$ is omitted and is by default the ground set of unlabeled instances $\mathcal{X}$.

[3] We refer the reader to the original paper [KSZ19] for a more formal description of "teacher mapping".

[4]The Warmuth hypothesis class is the smallest concept class for which RTD exceeds VCD.

[5] When $\mathsf{VCD}(H, \Psi_H) = 0$, this implies $|H| = 1$.

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
