[Supplementary Material]

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

| $x$ / $h$ | $x_1$ | $x_2$ | $x_3$ | $x_4$ | $x_5$ | $\mathcal{S}_{\text{const}} = \mathcal{S}_{\text{global}}$ | $\mathcal{S}_{\text{gvs}}$ | $\mathcal{S}_{\text{local}}$ | $\mathcal{S}_{\text{lvs}}$ |
|---|---|---|---|---|---|---|---|---|---|
| $h_1$ | 1 | 1 | 0 | 0 | 0 | $(x_1, x_2, x_4)$ | $(x_1, x_2)$ | $(x_1)$ | $(x_1)$ |
| $h_2$ | 0 | 1 | 1 | 0 | 0 | $(x_2, x_3, x_5)$ | $(x_2, x_3)$ | $(x_3)$ | $(x_2)$ |
| $h_3$ | 0 | 0 | 1 | 1 | 0 | $(x_1, x_3, x_4)$ | $(x_3, x_4)$ | $(x_3, x_4)$ | $(x_3)$ |
| $h_4$ | 0 | 0 | 0 | 1 | 1 | $(x_2, x_4, x_5)$ | $(x_4, x_5)$ | $(x_5, x_4)$ | $(x_4)$ |
| $h_5$ | 1 | 0 | 0 | 0 | 1 | $(x_1, x_3, x_5)$ | $(x_1, x_5)$ | $(x_5)$ | $(x_5)$ |
| $h_6$ | 1 | 1 | 0 | 1 | 0 | $(x_1, x_2, x_4)$ | $(x_2, x_4)$ | $(x_4)$ | $(x_3)$ |
| $h_7$ | 0 | 1 | 1 | 0 | 1 | $(x_2, x_3, x_5)$ | $(x_3, x_5)$ | $(x_3, x_5)$ | $(x_4)$ |
| $h_8$ | 1 | 0 | 1 | 1 | 0 | $(x_1, x_3, x_4)$ | $(x_1, x_4)$ | $(x_4, x_3)$ | $(x_5)$ |
| $h_9$ | 0 | 1 | 0 | 1 | 1 | $(x_2, x_4, x_5)$ | $(x_2, x_5)$ | $(x_4, x_5)$ | $(x_1)$ |
| $h_{10}$ | 1 | 0 | 1 | 0 | 1 | $(x_1, x_3, x_5)$ | $(x_1, x_3)$ | $(x_5, x_3)$ | $(x_2)$ |

(a) The Warmuth hypothesis class and the corresponding teaching sequences (denoted by $\mathcal{S}$).

| $h'$ | $\forall h' \in H$ |
|---|---|
| $\sigma_{\text{const}}(h'; \cdot, \cdot)$ | 0 |
| $\sigma_{\text{global}}(h'; \cdot, \cdot)$ | |

(b) $\sigma_{\text{const}}$ and $\sigma_{\text{global}}$

| $h \backslash h'$ | $h_1$ | $h_2$ | $h_3$ | $h_4$ | $h_5$ | $h_6$ | $h_7$ | $h_8$ | $h_9$ | $h_{10}$ |
|---|---|---|---|---|---|---|---|---|---|---|
| $\sigma_{\text{local}}(h'; \cdot, h = h_1)$ | 0 | 2 | 4 | 4 | 2 | 1 | 3 | 3 | 3 | 3 |
| $\dots$ | | | | | | | | | | |

(c) $\sigma_{\text{local}}$ representing the Hamming distance between $h'$ and $h$.

| $h'$ | $h_1$ | $h_2$ | $\dots$ |
|---|---|---|---|
| $H$ | $\{h_1, h_6\}$ | $\{h_2, h_7\}$ | $\dots$ |
| | $\{h_1\}$ | $\{h_2\}$ | $\dots$ |
| $\sigma_{\text{gvs}}$ | 0 | 0 | $\dots$ |

(d) $\sigma_{\text{gvs}}(h'; H, \cdot)$

| $h'$ | $h_1$ | $h_2$ | | $\dots$ |
|---|---|---|---|---|

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

# A  Supplementary Materials for §4

## A.1  An Example Hypothesis Class and the Teaching Sequences for the Batch Models

In this section, we provide an example hypothesis class where $\Sigma_{\text{const}}\text{-TD} = \text{TD} = 3$, $\Sigma_{\text{global}}\text{-TD} = \text{RTD} = 2$, and $\Sigma_{\text{gvs}}\text{-TD} = \text{NCTD} = 1$. The hypothesis class is specified in Table 3a. The preference functions inducing the optimal teaching sets for the examples are specified in Table 3b, 3c, and 3d.

| $\mathcal{X}$ / $\mathcal{H}$ | $x_1$ | $x_2$ | $x_3$ | $x_4$ | $x_5$ | $x_6$ | $\mathcal{S}_{\text{const}}$ | $\mathcal{S}_{\text{global}}$ | $\mathcal{S}_{\text{gvs}}$ |
|---|---|---|---|---|---|---|---|---|---|
| $h_1$ | 1 | 0 | 0 | 0 | 0 | 1 | $(x_1, x_6)$ | $(x_1, x_6)$ | $(x_1)$ |
| $h_2$ | 0 | 1 | 0 | 0 | 0 | 1 | $(x_2, x_6)$ | $(x_2, x_6)$ | $(x_2)$ |
| $h_3$ | 1 | 1 | 1 | 0 | 0 | 0 | $(x_3, x_4, x_5)$ | $(x_1)$ | $(x_3)$ |
| $h_4$ | 1 | 1 | 1 | 1 | 0 | 0 | $(x_4, x_5)$ | $(x_4, x_5)$ | $(x_4)$ |
| $h_5$ | 1 | 1 | 1 | 0 | 1 | 0 | $(x_4, x_5)$ | $(x_4, x_5)$ | $(x_5)$ |
| $h_6$ | 0 | 0 | 0 | 1 | 1 | 1 | $(x_4, x_5)$ | $(x_4, x_5)$ | $(x_6)$ |

(a) An example hypothesis class with the optimal teaching sets under different families of preference functions.

| $h'$ | $h_1$ | $h_2$ | $h_3$ | $h_4$ | $h_5$ | $h_6$ |
|---|---|---|---|---|---|---|
| $\sigma_{\text{const}}(h'; \cdot, \cdot)$ | 0 | 0 | 0 | 0 | 0 | 0 |

(b) Preference function $\sigma_{\text{const}}$

| $h'$ | $h_1$ | $h_2$ | $h_3$ | $h_4$ | $h_5$ | $h_6$ |
|---|---|---|---|---|---|---|
| $\sigma_{\text{global}}(h'; \cdot, \cdot)$ | 1 | 1 | 0 | 1 | 1 | 1 |

(c) Preference function $\sigma_{\text{global}}$

| $h'$ | $h_1$ | $h_2$ | $h_3$ | $h_4$ | $h_5$ | $h_6$ |
|---|---|---|---|---|---|---|
| $H$ | $\{h_1, h_3, h_4, h_5\}$ $\{h_1, h_3, h_4\}$ $\{h_1, h_3, h_5\}$ $\{h_1\}$ | $\{h_2, h_3, h_4, h_5\}$ $\{h_2, h_3, h_4\}$ $\{h_2, h_3, h_5\}$ $\{h_2\}$ | $\{h_3, h_4, h_5\}$ $\{h_3, h_4\}$ $\{h_3, h_5\}$ $\{h_3\}$ | $\{h_4, h_6\}$ $\{h_4\}$ | $\{h_5, h_6\}$ $\{h_5\}$ | $\{h_1, h_2, h_6\}$ $\{h_1, h_6\}$ $\{h_2, h_6\}$ $\{h_6\}$ |
| $\sigma_{\text{gvs}}(h'; H, \cdot)$ | 0 | 0 | 0 | 0 | 0 | 0 |

(d) Preference function $\sigma_{\text{gvs}}$. For all other $h', H$ pairs not specified in the table, $\sigma(h', H, \cdot) = 1$.

Table 3: An example hypothesis class where $\Sigma_{\text{const}}\text{-TD} = 3$, $\Sigma_{\text{global}}\text{-TD} = 2$, and $\Sigma_{\text{gvs}}\text{-TD} = 1$.

## A.2  Proof of Theorem 1

Before we prove our main results for the batch models, we first establish the following results on the non-clashing teaching. The notion of a non-clashing teacher was first introduced by [KKW07]. Our proof is inspired by [KSZ19] which shows the non-clashing property for collusion-free teacher-learner pair, under the batch setting.

**Lemma 5** *Assume $\sigma \in \Sigma_{\text{gvs}}$ is collusion-free. Then teacher $\text{T}$ must be non-clashing on $\mathcal{H}$. i.e., for any two distinct $h, h' \in \mathcal{H}$ such that $\text{T}(h)$ is consistent with $h'$, $\text{T}(h')$ cannot be consistent with $h$.*

**Proof** [Proof of Lemma 5] By definition of the preference function, we have $\forall \sigma \in \Sigma_{\text{gvs}}, h' \in \mathcal{H}$, $\sigma(h'; \mathcal{H}(Z'), \cdot) = g_\sigma(h', \mathcal{H}(Z'))$ for some function $g_\sigma$.

We then prove the lemma by contradiction. Assume that the teacher mapping $\text{T}$ isn't non-clashing. There exists $h \neq h' \in \mathcal{H}$, where $Z = \text{T}(h)$, and $Z' = \text{T}(h')$ are consistent with both, $h$ and $h'$.

Assume that the last current hypothesis before the teacher provides the last example of $Z$ is $h_1$. Then,

$$h = \underset{h'' \in \mathcal{H}(Z)}{\arg\min} \sigma(h''; \mathcal{H}(Z), h_1) = \underset{h'' \in \mathcal{H}(Z \cup Z')}{\arg\min} \sigma(h''; \mathcal{H}(Z \cup Z'), h_1) = \underset{h'' \in \mathcal{H}(Z \cup Z')}{\arg\min} g_\sigma(h'', \mathcal{H}(Z \cup Z')).$$

Where first equality is the definition of a teaching sequence. The second equality is by the definition of collusion-free preference (Definition 1). Similarly we have

$$h' = \underset{h'' \in \mathcal{H}(Z' \cup Z)}{\arg\min} g_\sigma(h'', \mathcal{H}(Z' \cup Z)).$$

Consequently, $h = h'$, which is a contradiction. This indicates that $\text{T}$ is non-clashing. ∎

Now we are ready to provide the proof for Theorem 1. We divide the proof of the Theorem 1 into three parts, each corresponding to the equivalence results for a different preference function family.

**Proof** [Proof of Theorem 1] Part 1 (reduction to $\mathsf{TD}$) and Part 2 (reduction to $\mathsf{RTD}$) of the proof are included in the main paper.

To establish the equivalence between $\Sigma_{\mathsf{gvs}}\text{-}\mathsf{TD}$ and $\mathsf{NCTD}$, we aim to show that for any hypotheses space $\mathcal{H}$, it holds (i) $\Sigma_{\mathsf{gvs}}\text{-}\mathsf{TD}_{\mathcal{X},\mathcal{H},h_0} \geqslant \mathsf{NCTD}(\mathcal{H})$, and (ii) $\Sigma_{\mathsf{gvs}}\text{-}\mathsf{TD}_{\mathcal{X},\mathcal{H},h_0} \leqslant \mathsf{NCTD}(\mathcal{H})$.

We first prove (i). According to Lemma 5, for any $\sigma \in \Sigma_{\mathsf{gvs}}$, a successful teacher T with $\sigma$ is non-clashing on $\mathcal{H}$. Therefore, $\Sigma_{\mathsf{gvs}}\text{-}\mathsf{TD}_{\mathcal{X},\mathcal{H},h_0} = \min_{\text{Successful Teacher T}} \max_{h\in\mathcal{H}} |\mathrm{T}(h)| \geqslant \min_{\text{Non-clashing teacher T}} \max_{h\in\mathcal{H}} |\mathrm{T}(h)| = \mathsf{NCTD}(\mathcal{H})$.

We now proceed to prove (ii). Consider any non-clashing teacher mapping T. First we will prove that there exists $\sigma \in \Sigma_{\mathsf{gvs}}$ such that $(\mathrm{T}, L_\sigma)$ is successful on $\mathcal{H}$. Here $L_\sigma$ is a learner corresponded to $\sigma$ as described in §2, and by "successful" we mean that the learner successfully outputs the target hypothesis when teaching terminates. In the following, we construct a preference function $\sigma$. First initialize $\sigma(\cdot; \cdot, \cdot) = 1$. Then, for every $h \in \mathcal{H}$, and every $S'$ which $\mathrm{T}(h) \subseteq S'$ and $S'$ is consistent with $h$ assign $\sigma(h; \mathcal{H}(S'), \cdot) = 0$.

We then prove (ii) by contradiction. Consider any set of examples $S$, and assume there exists two $h' \neq h' \in \mathcal{H}$ where $\sigma(h; \mathcal{H}(S), \cdot) = \sigma(h'; \mathcal{H}(S), \cdot) = 0$. Then $\mathrm{T}(h) \subseteq \mathcal{H}(S)$ and $\mathrm{T}(h') \subseteq \mathcal{H}(S)$, also $S$ is consistent with both $h$ and $h'$. This indicates that, both $\mathrm{T}(h)$ and $\mathrm{T}(h')$ must be consistent with both $h$, and $h'$. This contradicts with $T$ being non-clashing. Therefore, for every $h$, and $S'$ where $S'$ is consistent with $h$ and $\mathrm{T}(h) \subseteq S'$, and $h' \neq h$, we have $\sigma(h; \mathcal{H}(S'), \cdot) < \sigma(h'; \mathcal{H}(S'), \cdot)$. Consequently, after providing the examples $\mathrm{T}(h)$ to the learner $L_\sigma$, the learner will stay on $h$ even if she receives more consistent labeled examples. Therefore, $(\mathrm{T}, L_\sigma)$ is both collusion-free and successful on $\mathcal{H}$.

Therefore, we conclude that for any teacher mapping $T$ induced by $\sigma \in \Sigma_{\mathsf{gvs}}$, $\max_{h\in\mathcal{H}} |\mathrm{T}(h)| \geqslant \mathsf{TD}_{\mathcal{X},\mathcal{H},h_0}(\sigma)$. Consequently, $\Sigma_{\mathsf{gvs}}\text{-}\mathsf{TD}_{\mathcal{X},\mathcal{H},h_0} \leqslant \mathsf{NCTD}(\mathcal{H})$. Combining this results with (i) hence completes the proof. ∎

# B  Supplementary Materials for §5: Extension of Table 2

This section provides the details of preference functions for the Warmuth class.

| $\mathcal{H}$ \ $\mathcal{X}$ | $x_1$ | $x_2$ | $x_3$ | $x_4$ | $x_5$ | $\mathcal{S}_{\text{const}} = \mathcal{S}_{\text{global}}$ | $\mathcal{S}_{\text{gvs}}$ | $\mathcal{S}_{\text{local}}$ | $\mathcal{S}_{\text{lvs}}$ |
|---|---|---|---|---|---|---|---|---|---|
| $h_1$ | 1 | 1 | 0 | 0 | 0 | $(x_1, x_2, x_4)$ | $(x_1, x_2)$ | $(x_1)$ | $(x_1)$ |
| $h_2$ | 0 | 1 | 1 | 0 | 0 | $(x_2, x_3, x_5)$ | $(x_2, x_3)$ | $(x_3)$ | $(x_2)$ |
| $h_3$ | 0 | 0 | 1 | 1 | 0 | $(x_1, x_3, x_4)$ | $(x_3, x_4)$ | $(x_3, x_4)$ | $(x_3)$ |
| $h_4$ | 0 | 0 | 0 | 1 | 1 | $(x_2, x_4, x_5)$ | $(x_4, x_5)$ | $(x_5, x_4)$ | $(x_4)$ |
| $h_5$ | 1 | 0 | 0 | 0 | 1 | $(x_1, x_3, x_5)$ | $(x_1, x_5)$ | $(x_5)$ | $(x_5)$ |
| $h_6$ | 1 | 1 | 0 | 1 | 0 | $(x_1, x_2, x_4)$ | $(x_2, x_4)$ | $(x_4)$ | $(x_3)$ |
| $h_7$ | 0 | 1 | 1 | 0 | 1 | $(x_2, x_3, x_5)$ | $(x_3, x_5)$ | $(x_3, x_5)$ | $(x_4)$ |
| $h_8$ | 1 | 0 | 1 | 1 | 0 | $(x_1, x_3, x_4)$ | $(x_1, x_4)$ | $(x_4, x_3)$ | $(x_5)$ |
| $h_9$ | 0 | 1 | 0 | 1 | 1 | $(x_2, x_4, x_5)$ | $(x_2, x_5)$ | $(x_4, x_5)$ | $(x_1)$ |
| $h_{10}$ | 1 | 0 | 1 | 0 | 1 | $(x_1, x_3, x_5)$ | $(x_1, x_3)$ | $(x_5, x_3)$ | $(x_2)$ |

(a) The Warmuth hypothesis class and the corresponding teaching sequences (denoted by $\mathcal{S}$).

| $h'$ | $\forall h' \in H$ |
|---|---|
| $\sigma_{\text{const}}(h', \cdot, \cdot)$ | 0 |
| $\sigma_{\text{global}}(h', \cdot, \cdot)$ | |

(b) $\sigma_{\text{const}}$ and $\sigma_{\text{global}}$

| $h \backslash h'$ | $h_1$ | $h_2$ | $h_3$ | $h_4$ | $h_5$ | $h_6$ | $h_7$ | $h_8$ | $h_9$ | $h_{10}$ |
|---|---|---|---|---|---|---|---|---|---|---|
| $\sigma_{\text{local}}(h'; \cdot, h = h_1)$ | 0 | 2 | 4 | 4 | 2 | 1 | 3 | 3 | 3 | 3 |
| $\ldots$ | | | | | | | | | | |

(c) $\sigma_{\text{local}}$ representing the Hamming distance between $h'$ and $h$.

| $h'$ | $h_1$ | $h_2$ | $h_3$ | $h_4$ | $h_5$ | $h_6$ | $h_7$ | $h_8$ | $h_9$ | $h_{10}$ |
|---|---|---|---|---|---|---|---|---|---|---|
| $H$ | $\{h_1, h_6\}$ $\{h_1\}$ | $\{h_2, h_7\}$ $\{h_2\}$ | $\{h_3, h_8\}$ $\{h_3\}$ | $\{h_4, h_9\}$ $\{h_4\}$ | $\{h_5, h_{10}\}$ $\{h_5\}$ | $\{h_6, h_9\}$ $\{h_6\}$ | $\{h_7, h_{10}\}$ $\{h_7\}$ | $\{h_8, h_6\}$ $\{h_8\}$ | $\{h_9, h_7\}$ $\{h_9\}$ | $\{h_{10}, h_8\}$ $\{h_{10}\}$ |
| $\sigma_{\text{gvs}}$ | 0 | 0 | 0 | 0 | 0 | 0 | 0 | 0 | 0 | 0 |

(d) $\sigma_{\text{gvs}}(h'; H, \cdot)$

| $h'$ | $h_1$ | | $h_2$ | | $h_3$ | | $h_4$ | | $h_5$ | |
|---|---|---|---|---|---|---|---|---|---|---|
| $H$ | $\{h_1\} \cup$ $\{h_5, h_6, h_8, h_{10}\}^*$ | | $\{h_2\} \cup$ $\{h_1, h_7, h_6, h_9\}^*$ | | $\{h_3\} \cup$ $\{h_2, h_7, h_8, h_{10}\}^*$ | | $\{h_4\} \cup$ $\{h_3, h_6, h_8, h_9\}^*$ | | $\{h_5\} \cup$ $\{h_4, h_7, h_9, h_{10}\}^*$ | |
| $h$ | $h_1$ | | $h_1$ | $h_2$ | $h_1$ | $h_3$ | $h_1$ | $h_4$ | $h_1$ | $h_5$ |
| $\sigma_{\text{lvs}}$ | 0 | | 0 | 0 | 0 | 0 | 0 | 0 | 0 | 0 |

$$\vdots$$

| $h'$ | $h_6$ | | $h_7$ | | $h_8$ | | $h_9$ | | $h_{10}$ | |
|---|---|---|---|---|---|---|---|---|---|---|
| $H$ | $\{h_6\} \cup$ $\{h_1, h_4, h_5, h_9\}^*$ | | $\{h_7\} \cup$ $\{h_1, h_2, h_5, h_{10}\}^*$ | | $\{h_8\} \cup$ $\{h_1, h_2, h_3, h_6\}^*$ | | $\{h_9\} \cup$ $\{h_2, h_3, h_4, h_7\}^*$ | | $\{h_{10}\} \cup$ $\{h_3, h_4, h_5, h_8\}^*$ | |
| $h$ | $h_1$ | $h_6$ | $h_1$ | $h_7$ | $h_1$ | $h_8$ | $h_1$ | $h_9$ | $h_1$ | $h_{10}$ |
| $\sigma_{\text{lvs}}$ | 0 | 0 | 0 | 0 | 0 | 0 | 0 | 0 | 0 | 0 |

(e) $\sigma_{\text{lvs}}(h'; H, h)$. Here, $\{\cdot\}^*$ denotes all subsets.

Table 4: Teaching sequences with different preference functions for the Warmuth hypothesis class [DFSZ14]

## C  Supplementary Materials for §5: Proof for Theorem 2

We divide the proof into three parts. The first part shows that the interactions of the two families is $\Sigma_{\text{global}}$. In part 2 and part 3 of the proof, we show that there exist examples of hypothesis classes, such that $\Sigma_{\text{local}}\text{-TD}_{\mathcal{X},\mathcal{H},h_0} > \Sigma_{\text{gvs}}\text{-TD}_{\mathcal{X},\mathcal{H},h_0}$, or $\Sigma_{\text{local}}\text{-TD}_{\mathcal{X},\mathcal{H},h_0} < \Sigma_{\text{gvs}}\text{-TD}_{\mathcal{X},\mathcal{H},h_0}$.

### C.1  Part 1

In this subsection, we provide the full proof for part 1 of Theorem 2, i.e., $\Sigma_{\text{gvs}} \cap \Sigma_{\text{local}} = \Sigma_{\text{global}}$.

Intuitively, observe that the input domains between $\sigma_{\text{local}} \in \Sigma_{\text{global}}$ and $\sigma_{\text{gvs}} \in \Sigma_{\text{gvs}}$ overlaps at the domain of the first argument, which is the one taken by $\sigma_{\text{global}}$. Therefore, $\forall \sigma \in \Sigma_{\text{global}}, \sigma \in \Sigma_{\text{gvs}} \cap \Sigma_{\text{local}}$. We formalize such idea in the proof below.

**Proof** Assume $\sigma \in \Sigma_{\text{local}} \cap \Sigma_{\text{gvs}}$. Then, by the definitions of $\Sigma_{\text{local}}$ and $\Sigma_{\text{gvs}}$, we get

  (i) $\exists g^1$, s.t. $\forall h, h' \in \mathcal{H} : \sigma(h'; \cdot, h) = g^1(h', h)$, and
  (ii) $\exists g^2$, s.t. $\forall h' \in \mathcal{H}, H \subseteq \mathcal{H} : \sigma(h'; H, \cdot) = g^2(h', H)$

Now consider $h', h^1, h^2 \in \mathcal{H}$, and $H^1, H^2 \subseteq \mathcal{H}$. According to (i), $\sigma(h'; H^1, h^1) = \sigma(h'; H^2, h^1)$. Also, according to (ii) $\sigma(h'; H^2, h^1) = \sigma(h'; H^2, h^2)$. This indicates that, $\forall h', h^1, h^2 \in \mathcal{H}; H^1, H^2 \subseteq \mathcal{H} : \sigma(h'; H^1, h^1) = \sigma(h'; H^2, h^2)$. In other words, there exist $g^3 : \mathcal{H} \to \mathbb{R}$, such that $\forall h' \in \mathcal{H} : \sigma(h'; \cdot, \cdot) = g^3(h')$. Thus, $\sigma \in \Sigma_{\text{global}}$. ∎

### C.2  Part 2

**Part 2.** Next, we show that there exists $(\mathcal{H}, \mathcal{X})$, such that $\forall h_0 \in \mathcal{H}$, $\Sigma_{\text{local}}\text{-TD}_{\mathcal{X},\mathcal{H},h_0} > \Sigma_{\text{gvs}}\text{-TD}_{\mathcal{X},\mathcal{H},h_0}$. To prove this statement, we first establish the following lemma.

**Lemma 6** *For any $\mathcal{H}$, $\mathcal{X}$, and $h_0 \in \mathcal{H}$, if $\Sigma_{\text{local}}\text{-TD}_{\mathcal{X},\mathcal{H},h_0} = 1$, then $\Sigma_{\text{global}}\text{-TD}_{\mathcal{X},\mathcal{H},h_0} = 1$.*

**Proof** [Proof of Lemma 6] If $\Sigma_{\text{local}}\text{-TD}_{\mathcal{X},\mathcal{H},h_0} = 1$, there should be some $\sigma_{\text{local}} \in \Sigma_{\text{local}}$, such that $\text{TD}_{\mathcal{X},\mathcal{H},h_0}(\sigma_{\text{local}}) = 1$. Now consider $\sigma_{\text{global}}$ such that $\forall h', \sigma_{\text{global}}(h'; \cdot, \cdot) = \sigma_{\text{local}}(h'; \cdot, h_0)$. If $T_{\sigma_{\text{local}}}$ is the best teacher for $\sigma_{\text{local}}$, then $\forall h \in \mathcal{H} : |T_{\sigma_{\text{local}}}(h)| = 1$, this indicates that $h = \arg\min_{h' \in \mathcal{H}(T_{\sigma_{\text{local}}}(h))} \sigma_{\text{local}}(h'; \cdot, h_0)$ and $|\arg\min_{h' \in \mathcal{H}(T_{\sigma_{\text{local}}}(h))} \sigma_{\text{local}}(h'; \cdot, h_0)| = 1$. Subsequently, $h = \arg\min_{h' \in \mathcal{H}(T_{\sigma_{\text{local}}}(h))} \sigma_{\text{global}}(h'; \cdot, \cdot)$ and $|\arg\min_{h' \in \mathcal{H}(T_{\sigma_{local}})} \sigma_{\text{global}}(h'; \cdot, \cdot)| = 1$. In other words, $T_{\sigma_{\text{local}}}$ is also a teacher for $\sigma_{\text{local}}$. This indicates that, $\text{RTD}(\mathcal{H}) = \Sigma_{\text{global}}\text{-TD}_{\mathcal{X},\mathcal{H},h_0} = \text{TD}_{\mathcal{X},\mathcal{H},h_0}(\sigma_{\text{global}}) = 1$. ∎

Now we are ready to provide the proof for Part 2.

**Proof** [Proof of Part 2 of Theorem 2] We identify $\mathcal{H}$, $\mathcal{X}$, $h_0$, where $\Sigma_{\text{gvs}}\text{-TD}_{\mathcal{X},\mathcal{H},h_0} = 1$ and $\Sigma_{\text{global}}\text{-TD}_{\mathcal{X},\mathcal{H},h_0} = \text{RTD} = 2$. Table 3 illustrates such an example. In the example, since $\text{RTD} = 2$, then by Lemma 6, it must hold that $\Sigma_{\text{local}}\text{-TD}_{\mathcal{X},\mathcal{H},h_0} > 1 = \Sigma_{\text{gvs}}\text{-TD}_{\mathcal{X},\mathcal{H},h_0} = \text{NCTD}$. ∎

### C.3  Part 3

Here, we show that there exists a problem instance $(\mathcal{H}, \mathcal{X})$, such that $\forall h_0 \in \mathcal{H}$, $\Sigma_{\text{local}}\text{-TD}_{\mathcal{X},\mathcal{H},h_0} < \Sigma_{\text{gvs}}\text{-TD}_{\mathcal{X},\mathcal{H},h_0}$. Consider the hypothesis class which consists of the powerset $\mathcal{H} = \{0, 1\}^k$. First, as proven in Lemma 7 below, we show that $\forall h_0 \in \mathcal{H}$, $\Sigma_{\text{gvs}}\text{-TD}_{\mathcal{X},\mathcal{H},h_0} = \text{NCTD} \geqslant \lceil k/2 \rceil$.

**Lemma 7 (Based on Theorem 23 of [KSZ19])** *Consider the hypothesis class which consists of the powerset $\mathcal{H} = \{0, 1\}^k$. Then, NCTD $\geqslant \lceil k/2 \rceil$.*

**Proof** First we make the following observation: If $T$ is a non clashing teacher and $h, h' \in \mathcal{H}$ where $h = h' \triangle x$ (i.e., these two hypotheses only differ in their label on one instance), it must be the

case that $(x, h(x)) \in T(h)$, or $(x, h'(x)) \in T(h')$. This holds by nothing that since $h$, and $h'$ are only different on $x$, if $x$ is absent in their teaching sequences, this would lead to violation of the non-clashing property of the teacher.

Next we apply this observation on the powerset $k$ hypotheses class where $\mathcal{H}$ consists of all hypotheses which have length $k$. This indicates that for every $h \in \mathcal{H}$, and $0 \leqslant j \leqslant (k-1)$ all k variants $h \triangle x_j \in \mathcal{H}$. For all $0 \leqslant j \leqslant (k-1)$ by using the above observation, for a pair $h$ and $h \triangle x_j$, we drive $\sum_{i=0}^{2^k-1} |T(h_i)| \geqslant \frac{k \cdot 2^k}{2}$. By applying the pigeon-hole principle, this indicates that there exist an $h \in \mathcal{H}$, where $|T(h)| \geqslant \frac{k}{2}$. In other words $\mathsf{NCTD}(\mathcal{H}) \geqslant \lceil \frac{k}{2} \rceil$. ∎

Fix $k = 7$, we get $\Sigma_{\mathsf{gvs}}\text{-}\mathsf{TD}_{\mathcal{X},\mathcal{H},h_0} = \mathsf{NCTD}(\mathcal{H}) \geqslant 4$. On the other hand, we construct a preference function $\sigma \in \Sigma_{\mathsf{local}}$, where $\Sigma_{\mathsf{local}}\text{-}\mathsf{TD}_{\mathcal{X},\mathcal{H},h_0} \leqslant \mathsf{TD}_{\mathcal{X},\mathcal{H},h_0}(\sigma) = 3$ for $k = 7$.

The example is detailed in Figure 5. Intuitively, for any $h_0 \in \mathcal{H}$, we construct a tree of hypotheses with branching factor 7 at the top level, branching factor of 6 at the next level, and so on. Here, each branch corresponds to one teaching example, and each path from $h_0$ to $h \in \mathcal{H}$ corresponds to a teaching sequence $\mathsf{T}_{\mathsf{local}}(h)$. We need a tree of depth at most 3 to include all the $2^7 = 128$ hypotheses to be taught as nodes in the tree. This gives us a constructive procedure of $\sigma$ functions achieving $\mathsf{TD}_{\mathcal{X},\mathcal{H},h_0}(\sigma) = 3 < \Sigma_{\mathsf{gvs}}\text{-}\mathsf{TD}_{\mathcal{X},\mathcal{H},h_0}$, which completes the proof.

| $h$ | $\mathcal{X}$ | Preference Function $\sigma(.;h)$ | Teaching Sequence |
|---|---|---|---|
| $h_0$ | 0 0 0 0 0 0 0 | $h_0 > h_1 > h_2 > h_3 > h_4 > h_5 > h_6 > h_7 >$ others | $((x_0, 0))$ |
| $h_1$ | 1 0 0 0 0 0 0 | $h_1 > h_8 > h_9 > h_{10} > h_{11} > h_{12} > h_{13}$ others | $((x_0, 1))$ |
| $h_8$ | 1 1 0 0 0 0 0 | $h_8 > h_{44} > h_{45} > h_{46} > h_{47} > h_{48} >$ others | $((x_0, 1), (x_1, 1))$ |
| $h_9$ | 1 1 1 0 0 0 0 | $h_9 > h_{79} > h_{80} > h_{81} > h_{82} > h_{83} >$ others | $((x_0, 1), (x_2, 1))$ |
| $h_{10}$ | 1 1 1 1 0 0 0 | $h_{10} > h_{114} > h_{115} >$ others | $((x_0, 1), (x_3, 1))$ |
| $h_{11}$ | 1 1 1 1 1 0 0 | $h_{11} >$ others | $((x_0, 1), (x_4, 1))$ |
| $h_{12}$ | 1 1 1 1 1 1 0 | $h_{12} >$ others | $((x_0, 1), (x_5, 1))$ |
| $h_{13}$ | 1 1 1 1 1 1 1 | $h_{13} >$ others | $((x_0, 1), (x_6, 1))$ |
| $h_{44}$ | 1 1 0 1 0 0 0 | $h_{44} >$ others | $((x_0, 1), (x_1, 1), (x_3, 1))$ |
| $h_{45}$ | 1 1 0 1 1 0 0 | $h_{45} >$ others | $((x_0, 1), (x_1, 1), (x_4, 1))$ |
| $h_{46}$ | 1 1 1 0 1 0 0 | $h_{46} >$ others | $((x_0, 1), (x_1, 1), (x_2, 1))$ |
| $h_{47}$ | 1 1 0 0 0 1 0 | $h_{47} >$ others | $((x_0, 1), (x_1, 1), (x_5, 1))$ |
| $h_{48}$ | 1 1 0 0 1 0 1 | $h_{48} >$ others | $((x_0, 1), (x_1, 1), (x_6, 1))$ |
| $h_{79}$ | 1 0 1 0 0 0 0 | $h_{79} >$ others | $((x_0, 1), (x_2, 1), (x_1, 0))$ |
| $h_{80}$ | 1 0 1 0 1 0 0 | $h_{80} >$ others | $((x_0, 1), (x_2, 1), (x_4, 1))$ |
| $h_{81}$ | 1 0 1 0 1 1 0 | $h_{81} >$ others | $((x_0, 1), (x_2, 1), (x_5, 1))$ |
| $h_{82}$ | 1 1 1 1 0 1 0 | $h_{47} >$ others | $((x_0, 1), (x_2, 1), (x_3, 1))$ |
| $h_{83}$ | 1 0 1 1 1 0 1 | $h_{83} >$ others | $((x_0, 1), (x_2, 1), (x_6, 1))$ |
| $h_{114}$ | 1 0 0 1 0 0 0 | $h_{114} >$ others | $((x_0, 1), (x_3, 1), (x_1, 0))$ |
| $h_{115}$ | 1 0 0 1 1 0 0 | $h_{115} >$ others | $((x_0, 1), (x_3, 1), (x_4, 1))$ |

Table 5: More details about Figure 5: This table lists down all the hypotheses in the left branch of the tree. For each of these hypotheses, it shows the preference function from the hypothesis, as well as the teaching sequence to teach the hypothesis. Consider $h_9$: We have $\sigma(., h_9) = \{h_9 > h_{79} > h_{80} > h_{81} > h_{82} > h_{83} >$ others$\}$. Also, we have teaching sequence for $h_9$ as $\{(x_0, 1), (x_2, 1)\}$.

(a) This figure is representing the teaching sequence for first four for direct children of $h_0$ (top four most preferred hypothesis of $h_0$ after $h_0$) and all of their children.

(b) This figure is representing the teaching sequence for next three direct children of $h_0$ (next three most preferred hypothesis of $h_0$) and all of their children.

Figure 5: Details of teaching sequences, for a preference function $\sigma \in \Sigma_{\text{local}}$, where $\mathsf{TD}_{\mathcal{X},\mathcal{H},h_0}(\sigma) = 3$ for powerset $k = 7$ class. For any hypothesis the cell with blue color is representing last teaching example in teaching sequence, and the cells with red color are representing rest of teaching sequence. Also see Table 5 that lists down details for all the hypotheses in the left branch of the tree.

# D Supplementary Materials for §5.3

## D.1 Proof of Lemma 4

In this section, we extend the proof sketch of Lemma 4 in the main paper into the full proof. A useful notion for this proof is the notion of $H$-*distinguishable set*:

**Definition 7 (Based on [DFSZ14])** *A set of instances $X \subseteq \mathcal{X}$ is $H$-distinguishable, if $|H_{|X}| = |H|$.*

For completeness, we also incorporate part of the proof sketch from §5.3 into the extended proof below.

**Proof** [(Extended) Proof of Lemma 4] Let us define $H_x = \{h \in H : h \triangle x_{|\Psi_H} \in H_{|\Psi_H}\}$. Here, $h \triangle x$ denotes the hypothesis that only differs from $h$ on the label of $x$. Fix a reference hypothesis $h_H$. $\forall x_j \in \Psi_H$, let $y_j = 1 - h_H(x_j)$ be the opposite label of $x_j$ as provided by $h_H$. As highlighted in Line 9 of Algorithm 2, we consider the set $H^{y_1}_{x_1} = \{h \in H_{x_1} : h(x_1) = y_1\}$ as the first partition.

| $H$ \ $\Psi_H$ | $x_1$ | $x_2$ | ... | $x_{m-1}$ |
|---|---|---|---|---|
| $h_0$ | 0 | 00 ... 0 | | |
| $h_1$ | 0 | a | | |
| $h_2$ | 1 | b | | |
| $h_3$ | 0 | b | | |
| $h_4$ | 1 | c | | |
| $h_5$ | 1 | d | | |
| $h_6$ | 1 | e | | |
| $h_7$ | 1 | a | | |

(a) $H$

| $H_{x_1}$ \ $\Psi_H$ | $x_1$ | $x_2$ | ... | $x_{m-1}$ |
|---|---|---|---|---|
| $h_1$ | 0 | a | | |
| $h_2$ | 1 | b | | |
| $h_3$ | 0 | b | | |
| $h_7$ | 1 | a | | |

(b) $H_{x_1}$

| $H^1$ \ $\Psi_H$ | $x_1$ | $x_2$ | ... | $x_{m-1}$ |
|---|---|---|---|---|
| $h_2$ | 1 | b | | |
| $h_7$ | 1 | a | | |

(c) $H^1 = H^{y_1=1}_{x_1}$

Table 6: Illustrative example for constructing the first partition $H^1 = H^{y_1=1}_{x_1}$.

In Table 6, we provide an example hypothesis class where we show how to construct the first partition $H^{y_1}_{x_1}$. Table 6a shows the hypothesis class $\mathcal{H}$ (here $a \neq b \neq c \neq d \neq e$) and $h_{\mathcal{H}} = h_0$. Table 6b shows the resulting set of hypotheses $H_{x_1} = \{h \in H : h \triangle x_{1|\Psi_H} \in H_{|\Psi_H}\}$, and Table 6c shows the first partition $H^{y_1=1}_{x_1}$.

We denote $H^1 := H^{y_1}_{x_1}$, and define $\Psi_{H^1} \subseteq \Psi_H \backslash \{x_1\}$ to be any compact-distinguishable set on $H^1$.

**Lower VCD.** Let $d = \mathsf{VCD}(H, \Psi_H)$. In the following, we prove that $\mathsf{VCD}(H^1, \Psi_{H^1}) \leqslant d - 1$. We consider the following two cases:

1. If $d > 1$, then
   $$\mathsf{VCD}(H^1, \Psi_{H^1}) \leqslant \mathsf{VCD}(H^{y_1}_{x_1}, \Psi_H) = \mathsf{VCD}(H_{x_1}, \Psi_H) - 1 \leqslant \mathsf{VCD}(H, \Psi_H) - 1 \leqslant d - 1$$
   Since $\Psi_{H^1} \subset \Psi_H$, the first inequality is due to the monotonicity of $\mathsf{VCD}$. The equality follows from the fact that, for all $h \in H^{y_1}_{x_1}$, it holds that $h(x_1) = y_1$ and $h \triangle x_{1|\Psi_H} \in H_{x_1|\Psi_H}$. This indicates that, $X \subseteq \Psi_H$ shatters $H^{y_1}_{x_1}$, iff $X \cup \{x_1\}$ shatters $H_{x_1}$. The second inequality comes from the fact that $\mathsf{VCD}$ is monotonic.

2. If $d = 1$ and $|H^{y_1}_{x_1}| \geqslant 2$, then
   similar to the previous case we have the following: $\mathsf{VCD}(H_{x_1}, \Psi_H) \leqslant \mathsf{VCD}(H, \Psi_H) = 1$ and $\mathsf{VCD}(H_{x_1}, \Psi_H) = \mathsf{VCD}(H^{y_1}_{x_1}, \Psi_H) + 1$. Subsequently, $\mathsf{VCD}(H^1, \Psi_{H^1}) = 0$.

3. If $d = 1$ and $|H^{y_1}_{x_1}| < 2$, then
   since $|H^{y_1}_{x_1}| < 2$, by definition, we have $\mathsf{VCD}(H^1, \Psi_{H^1}) = 0$ and hence is less than $d = 1$.

That is, the first partition $H^1, \Psi_{H^1}$ has $\mathsf{VCD}(H^1, \Psi_{H^1}) \leqslant d - 1$, i.e., $H^1$ satisfies property (i). In addition, it is clear that $H^1_{|\{x_1\}} = \{y_1\} = \{1 - h_H(x_1)\}$. Therefore, $H^1$ also satisfies property (ii).

**Non-emptiness of $H^1$.** For the sake of contradiction assume that $H^1$ is empty. Note that $\Psi_H$ is $H$-distinguishable. Since $H^1$ is empty, this means that there is no pair of hypotheses that differ only on $x_1$. This in turn indicates that $\Psi_H \backslash \{x_1\}$ is $H$-distinguishable. However, $|\Psi_H \backslash \{x_1\}| < |\Psi_H|$ and this is in contradiction to the assumption that $\Psi_H$ is compact-distinguishable on $H$.

**Continuing to create partitions.** Next, we remove the first partition $H^1$ from $H$, and continue to create the above mentioned partitions on $H_{\text{rest}} = H \backslash H^1$ and $X_{\text{rest}} = \Psi_H \backslash \{x_1\}$. We claim that $H_{\text{rest}}, X_{\text{rest}}$ exhibit the following properties.

1. $X_{\text{rest}}$ is $H_{\text{rest}}$-distinguishable (see Definition 7).

   For the sake of contradiction, assume that there exists a pair of hypotheses $h^1, h^2 \in H_{\text{rest}}$ such that $h^1_{|X_{\text{rest}}} = h^2_{|X_{\text{rest}}}$. However, we know that $h^1_{|\Psi_H} \neq h^2_{|\Psi_H}$. Then, these two hypotheses should have been in $H_{x_1}$ and only one of them could have stayed in $H_{\text{rest}}$. Hence, there is no such pair of hypotheses in $H_{\text{rest}}$ and this completes the proof of the statement.

2. $X_{\text{rest}}$ is also a compact-distinguishable on $H_{\text{rest}}$.

   We now provide a concrete proof for the above statement. Imagine $X \subseteq X_{\text{rest}}$ is an $H_{\text{rest}}$-distinguishable set. In the following, we prove that $X \cup \{x_1\}$ is $H$-distinguishable.

   For the sake of contradiction assume that, $X \cup \{x_1\}$ isn't $H$-distinguishable. This indicates that there exist two hypotheses $h^1 \neq h^2 \in H$, where they are equal on $X \cup \{x_1\}$, i.e., $h^1_{|X \cup \{x_1\}} = h^2_{|X \cup \{x_1\}}$; also this implies $h^1_{|X} = h^2_{|X}$. Since $H = H_{\text{rest}} \cup H^1$, we consider the following three cases.

   (i) $h^1, h^2 \in H_{\text{rest}}$. Since $X$ is $H_{\text{rest}}$-distinguishable, it is a contradiction that $h^1_{|X} = h^2_{|X}$.

   (ii) $h^1, h^2 \in H^1$. By the construction of $H^1$, there exist $\hat{h}^1, \hat{h}^2 \in H_{\text{rest}}$, such that $\hat{h}^1_{|X \cup \{x_1\}} = h^1 \triangle x_{1|X \cup \{x_1\}}$ and $\hat{h}^2_{|X \cup \{x_1\}} = h^2 \triangle x_{1|X \cup \{x_1\}}$. Furthermore, since $h^1_{|X} = h^2_{|X}$, we must have $\hat{h}^1_{|X} = \hat{h}^2_{|X}$, which contradicts the fact that $X$ is $H_{\text{rest}}$-distinguishable.

   (iii) $h^1 \in H^1, h^2 \in H_{\text{rest}}$. By the construction of $H^1$, there exist $\hat{h}^1 \in H_{\text{rest}}$, such that $\hat{h}^1_{|X \cup \{x_1\}} = h^1 \triangle x_{1|X \cup \{x_1\}}$. Furthermore, since $h^1_{|X} = h^2_{|X}$, we must have $\hat{h}^1_{|X} = h^2_{|X}$, which contradicts the fact that $X$ is $H_{\text{rest}}$-distinguishable.

   Therefore, we conclude that $X \cup \{x_1\}$ is $H$-distinguishable. Recall that $\Psi_H$ is compact-distinguishable on $H$. This indicates that $\Psi_H = X \cup \{x_1\}$, and subsequently $X = X_{\text{rest}}$. This indicates that $X_{\text{rest}}$ is compact-distinguishable on $H_{\text{rest}}$.

3. If $|\Psi_H| > 1$, then $|H_{\text{rest}}| > 1$.

   We prove the above statement by contradiction. Assume that $|H_{\text{rest}}| = 1$. Since we know that $H^1$ is non empty, hence $|H_{\text{rest}}| = 1$ implies that $|H^1| = 1$. Let $H^1 = \{h\}$, and $H_{\text{rest}} = \{h'\}$, then $h'_{|\Psi_H} = h \triangle x_{1|\Psi_H}$. Since we know that $H = H^1 \cup H_{\text{rest}}$, subsequently $\{x_1\}$ is compact-distinguishable on $H$, which is in contradiction to the assumption that $\Psi_H$ is compact-distinguishable.

**Case of $|X_{\text{rest}}| > 1$.** Therefore, we can repeat the above procedure (Line 7– Line 10, Algorithm 2) to create the subsequent partitions. This process continues until the size of $X_{\text{rest}}$ reduces to 1, i.e. $X_{\text{rest}} = \{x_{m-1}\}$. Until then, we obtain partitions $\{H^1, ..., H^{m-2}\}$. By construction, $H^j$ satisfy properties (i) and (ii) for all $j \in [m-2]$.

Note that each step $X_{\text{rest}}$ is compact $H_{\text{rest}}$-distinguishable set. This implies that we have never lost a hypothesis in this process, i.e., all hypotheses in $H$ were either in one of $H_j$'s or in $H_{\text{rest}}$.

**Case of $|X_{\text{rest}}| = 1$.** It remains to show that the last two partitions $H^{m-1}$ and $H^m$ also satisfy properties (i) and (ii); additionally we need to satisfy property (iii). Since $X_{\text{rest}} = \{x_{m-1}\}$, and $|H_{\text{rest}}| > 1$ before we start iteration $m-1$, there must exist exactly two hypotheses in $H_{\text{rest}}$. Therefore $|H^{m-1}|, |H^m| = 1$, and $H^{m-1}_{|\{x_{m-1}\}} = \{\{y_{m-1}\}\}$. This implies that $\mathsf{VCD}(H^{m-1}, \Psi_{H^{m-1}}) = \mathsf{VCD}(H^m, \Psi_{H^m}) = 0 \leqslant d-1$. Furthermore, notice that for every $j \in [m-1], h \in H^j, h_H(x_j) \neq h(x_j)$. This indicates $h_H \in H_m$. Since $|H_m| = 1$, we get $H_m = \{h_H\}$ which completes the proof. ∎

## D.2 Supplementary Materials for the Proof of Theorem 3

Our proof of Theorem 3 in the main paper relies on the fact that every teaching example $(x_j, y_j)$, where $x_j \in \Psi_H$ and $y_j = 1 - h(x_j)$ for some fixed $h$, corresponds to a unique version space $V^j$. The proof depends on the following lemma.

**Lemma 8** *Fix $H \subseteq \mathcal{H}$, and let $\Psi_H \subseteq \mathcal{X}$ be a compact-distinguishable set on $H$. For any $x, x' \in \Psi_H$ and $y, y' \in \{0, 1\}$ such that $(x, y) \neq (x', y')$, the resulting version spaces $\{h \in H : h(x) = y\}$ and $\{h \in H : h(x') = y'\}$ are different.*

**Proof** [Proof of Lemma 8] Denote $A = \{h \in H : h(x) = y\}$ and $B = \{h \in H : h(x') = y'\}$. We consider the following two cases: (1) $y = y'$, and (2) $y \neq y'$. For the first case where $y = y'$, if $A = B$, this would violate the first part of property (i) of Lemma 4, (i.e., there do not exist distinct $x, x'$ s.t. $\forall h \in H : h(x) = h(x')$. For the second case, if $A = B$, this would violate the second part of property (i). Hence it completes the proof. ∎