[Reviews · NeurIPS 2019]

Reviewer 1



In this paper, a unified view of different teaching models is proposed based on preference functions. It shows that classical teaching models and their corresponding teaching complexity measures can be treated as teaching under preference functions with different levels of flexibility. Even though it is known in the literature that preference functions play a crucial role in teaching, proposing such unified view is still meaningful for inspiring future researches. Furthermore, the most interesting result in the paper is the study on the family of “local version space” preference function, which relies on both the current version space and the learner’s current local hypothesis. A constructive proof is provided to show that there exists such LVS preference function, under which the size of the optimal teaching set is linearly upper bounded by the VC dimension of the hypothesis space. This is an interesting result since studying the relationship between VC and the teaching dimension is one of the most important problem in machine teaching. As cracking the well-known open problem about the relationship between recursive teaching dimension, which corresponds to teaching with a global preference function, and VC dimension is still difficult, the above result reveals the importance of designing more flexible teaching models. To my understanding, this paper provides a potentially high impact to inspire further researches towards this direction. Further suggestions: 1) In the proof of Theorem 3, the notion of VC dimension w.r.t. a hypothesis class and a specific dataset is utilized with formal definition. This definition is different from classical VC dimension which involves only on the hypothesis class. Due to its importance in the proof, a formal definition is bettered to be included. 2) The room to improve paper presentation and organization is significant. To my understanding, the current version of the paper is not friendly enough to readers without a good knowledge about previous machine teaching literatures. For example, it is not discussed in detail about the teaching protocols: whether the teacher knows about the learner’s preference function? Will the teacher know the preference function beforehand, or get better knowledge during the sequential teaching process? A more detailed explanation about such stuffs and intuition behand the main results can be beneficial to further improve the potential impact of the paper. ------ after rebuttal: I have read all the reviews and the author response. I agree with other reviewers that the work is interesting and novel, thus my score remains the same.

Reviewer 2



Originality: As far as I know, such a sequential teaching model (where the preference function generally depends on both the learner's version space and current hypothesis) has never been considered in the literature. I feel it is an elegant idea; moreover, the fact that the general parameter \Sigma-TD_{\chi,\mathcal{H},h_0} reduces to the RTD/PBTD and NCTD for different families of preference functions is evidence, I think, that this paradigm is a natural generalisation of previously studied teaching models. The proofs combine combinatorial insights with known ideas in computational learning theory. The proof of the main lemma (Lemma 4), for example, defines a family of sets H_{x_i}^{y_i} that are very similar to the "reduction" C^{x} of a concept class C w.r.t. some instance x (see, for example, Section 6.1 and Theorem 9 in "Sample compression, learnability and the Vapnik-Chervonenkis dimension" by Sally Floyd and Manfred Warmuth, Machine Learning 21, 269--304, 1995). The fact that the VC-dimension of H^1 w.r.t. \Psi_H is at most d-1 follows almost immediately from the proof of the fact that the reduction C^{x} has VC-dimension at most d-1, where d = VCD(C) (Ibid.). As I see it, a new insight of the author(s) is to take a "compact-distinguishable" subset of the instance space in order to ensure that each set H^i is nonempty; another new insight is to recursively apply this construction so as to obtain a partition of the original concept class (using the fact that at each step of the recursion, the VCD of the next partition is strictly reduced). Quality: In general, I think the paper is technically sound. There are a few small issues regarding notation and definitions (please see "Minor Comments" below for details), all of which can probably be quickly fixed. All the claims in the paper are substantiated with complete proofs; I checked the main proofs in the appendix and they appear to be generally correct. Clarity: I could generally follow the definitions, theorems and main arguments in the proofs. I was a bit confused by certain aspects of Algorithm 2 at first, but after working through it, using the Warmuth class as an example, I think (at least I hope) I understand it better. Please refer to the comment/question below regarding Algorithm 2. There seem to be quite a few typos and grammatical/spelling errors (especially missing articles) throughout the paper, particularly in the supplementary material. These errors were not taken into account in the overall score; however, I strongly recommend that the whole paper, including the supplementary material, be thoroughly proofread again. Significance: That the author(s) found a (sequential) teaching complexity measure that is linear in the VCD, and in doing so unifying previously studied teaching models within a general paradigm, is in my opinion significant. It establishes a firm connection between the "hardness" of teaching and the VCD of any given finite concept class, complementing earlier similar results on the relation between the RTD and VCD (the current best upper bound on the RTD in terms of the VCD is a quadratic function). Reformulating the NCTD as a teaching parameter corresponding to a family of preference functions depending on the learner's version space might also open up a new approach to solving the open problem of whether the NCTD/RTD is linear in the VCD (though at this point I am not sure whether such an approach makes the problem any easier). Comment/question regarding Algorithm 2: After Lines 7 to 13 are executed for x_1,x_2,...,x_i in \Psi_H for some i < m-1, is the instance space X updated to X \setminus {x_1,...,x_i} ? In Line 8, h \triangle x is used in the definition of the set assigned to H_x^y. Based on the example of the Warmuth class, it seems that h \triangle x here denotes the hypothesis that only differs with h on the label of x with respect to the instance space X \setminus {x_1,...,x_i}, not with respect to the original instance space X. If this is indeed the case, then it might be helpful to include an update of the instance space explicitly between Lines 7 or 13, or emphasise in Line 8 that h \triangle x denotes the hypothesis that differs from h only on x when the instance space is X \setminus {x_1,...,x_i}. Minor Comments: 1. Page 1, line 17: Perhaps delete O(VCD) in this line, since it is already mentioned that the teaching complexity is linear in the VC dimension of the hypothesis class. 2. Page 3, lines 98 and 99: Perhaps write \mathcal{H}(z) as \mathcal{H}({z}) or mention that \mathcal{H}({z}) is written as \mathcal{H}(z) "for the sake of convenience". 3. Page 3, definition of U_{\sigma}(H,h,h^*): in the first condition, it looks like there is a type mismatch error; the left-hand side is a number but the right-hand side is a set. Should it be \exists z[C_{\sigma}(H,h,z) = {h^*}] ? 4. Page 4, lines 121 and 122: Explain what the vertical bars surrounding argmin... denote, or perhaps remove them. 5. Page 4, definitions of \Sigma_{const}, \Sigma_{global} and \Sigma_{gvs}: Should the existential quantifier precede the universal quantifier in each of these definitions? For example, in the definition of \Sigma_{const}, it seems that we need a single value c for all h', H and h. 6. Page 5, line 155: The term "teacher mapping" was not defined. 7. Page 5, line 164: The meaning of the notation VCD(\chi,\mathcal{H}), though perhaps not too hard to guess, could be mentioned in Section 2. 8. Page 5, line 159: Perhaps explain here why the Warmuth hypothesis class is interesting/significant, e.g. it is the smallest concept class for which RTD exceeds VCD. 9. Page 5, lines 176 and 179, definitions of \Sigma_{local} and \Sigma_{lvs}: Similar to point 5. 10. Page 6, line 196: Perhaps make clear whether or not H_{|X} is a class of sequences (since "pattern" seems to suggest that the order of the h(x_i)'s is used). 11. Page 6, line 201: "...does not contain any pair of [distinct] examples..." 12. Page 6, line 207: "...we can divide H [into]..." 13. Page 6, line 209: It looks like there should be another pair of braces surrounding {1-h_H(x_j)}. 14. Page 6, line 217: "...provide the detailed proof [in] the appendix" or replace "provide" by "defer". 15. Page 7, line 10 of Algorithm 2: I think the braces around x,y should be replaced with round brackets. 16. Page 8, lines 248-249: Missing article between "with" and "help". 17. Page 8, line 263: H_{j^*} -> H^{j^*} ? 18. Page 8, line 268: Missing article between "and" and "learner". 19. Page 8, line 280: Missing article between "have" and "win-stay, lose shift". (Also, it looks like "loose" should be replaced by "lose"...) 20. Page 8, line 297: family -> families 21. Page 12, line 385: Superfluous comma after the first word "that". 22. Page 12, line 390: Delete "of set" ? 23. Page 12, lines 395 and 399, and also at a few other places: I suggest writing "consequently" rather than "subsequently". 24. Page 17, line 478: hypothesis -> hypotheses 25. Page 17, line 486, and also at a few other places: "...is contradiction with..." -> "...is in contradiction to.." or "...contradicts the assumption that..." 26. Page 18, lines 519 and 521: H_{x_1}^{1-y_1} -> H_{x_1}^{y_1} ? 27. Page 18, line 534: H^{m-1}_{{x_{m-1}}} = y_{m-1} -> H^{m-1}{|{x_{m-1}}} = {{y_{m-1}}} ? (Missing vertical bar before x_{m-1} and missing braces around y_{m-1} ?) 28. Page 18, line 540: examples -> example 29. Page 18, line 541: V_j -> V^j ? 30. Page 19, line 549: Explain what z is. In the same line: "...there [do] not exist [distinct]..." 31. Page 19, line 557: Capitalise 's' in "since". In the same line: drive -> derive ? (I might have missed other typos or spelling/grammatical errors; please proofread again.) * Response to author feedback Thank you very much for the detailed feedback. I think this is a very good piece of work, and I will keep my current score.

Reviewer 3



The paper is on teaching models, a topic in computational learning theory. One of the recurring issues is how to formulate a model which avoids collusion, i.e., cheating'', where the teacher communicates the target to the learner using some encoding of its description into the teaching examples. The standard approach to teaching is through a set of examples uniquely defining the target; thus in this case the learner's task is to find a consistent hypothesis. More recent approaches, such as the recursive teaching dimension and the model of [CSMA+18], assume more sophisticated learners. In this paper an even more sophisticated class of learners is defined. Here learning is done in a sequential process, and in each step the learner finds a most preferred hypothesis, based on an evaluation of the current version space and the last hypothesis. This model is closely related to [CSMA+18], except here the version space is an additional parameter in the preference function. The results of the paper are interesting in themselves, and they also provide an interesting picture for the relative power of the various teaching models. The second half of Theorem 3 is an example of the former: it shows that in the most general model there is already a preference function which is proportional to the VC-dimension (whether this holds for RTD is a major open problem). This result is based on an interesting combinatorial lemma (Lemma 4). The latter is summarized in Figure 1 (where the relationships to previous models are given in Table 1). Comments The paper emphasizes the critical role of the appropriate definition of collusion-freeness for sequential models. The corresponding part of the proof of Theorem 3, however, is relegated to the Appendix. Some discussion of this should be given in the main text. Also, the concluding section mentions defining (other?) notions of collusion-freeness in sequential models. Here it should be clarified what are the issues with the current definition.

[Author Response · NeurIPS 2019]

We thank the reviewers for their valuable suggestions. Please find our answers (**A**) for each reviewer (**R**) below.

**R1, R2**: *Formal definition of* VCD

**A**: We will add the following definition: $\mathsf{VCD}(X, H) = \max |X'|$, s.t. $X' \subseteq X$ and $|H_{|X'}| = 2^{|X'|}$.

**R1**: *A more detailed explanation about the teaching models/protocol and intuition behind the main result*

**A**: As suggested by the reviewer, we will incorporate a detailed explanation of the existing teaching models and
protocols in the updated version of the paper. In particular, we will clarify that the teacher knows the learner's preference
function. This is the protocol used in existing teaching models for both the batch settings (e.g., as in RTD/PBTD
[ZLHZ11, GRSZ17]) and the sequential settings (e.g., as in [CSMA+18]).

**R2**: *Insights on the proof of the main lemma (Lemma 4) and connection to the reference*

**A**: Thanks for pointing out the similarity between the proof of our Lemma 4 and the proof of Theorem 9 in [FW95]. We
will acknowledge this connection and add a proper discussion in the revision. Concretely, the concept class $C^{\{x\}}$ of
[FW95] is equivalent to our definition of $H_x^0$ (line 221). [FW95] then applied induction on $C^{\{x\}}$ and $C - x$ (which
are represented as $H_x^0$, and $H \backslash H_x^0$ in our paper, respectively). As pointed out in the review, our novelty lies in that we
introduced (i) "compact-distinguishable" set to ensure that each $H^j$ is non-empty, and (ii) a recursive procedure for
constructing the preference function.

**R2**: *Questions regarding Algorithm 2*

**A**: We realized that there were some notation issues with Algorithm 2, and we agree with the fix suggested in the review.
We will incorporate the following updates which are related to Algorithm 2 as detailed below: In Algorithm 2, Line 8,
we should have $H_x^y \leftarrow \{h' \in H : h' \triangle x_{|X_{\text{rest}}} \in H_{|X_{\text{rest}}}, h'(x) = y\}$, where $X_{\text{rest}}$ (as described in Line 226–229) denotes
the set of instances in $\Psi_H$ that have not been traversed in the current for-loop. We will revise the algorithm accordingly
to make sure the notations are consistent and self-contained. Furthermore, we will also update the Appendix, in
particular, between Line 502–525, with proper conditions (e.g., among others, in Line 505, we will update $h' = h \triangle x_1$
into $h'_{|\Psi_H} = h \triangle x_{1|\Psi_H}$ to be more explicit about the instances being considered).

**R2**: *"Minor Comments": Typos, grammatical/spelling errors, and notation issues*

**A**: We greatly appreciate the time and effort spent by the reviewer in pointing us to the minor issues. We will thoroughly
proofread the paper and fix all the minor issues pointed out in the reviews. Also, we will address the following
definitions/notations as pointed out by the reviewer: [(3) *Page 3, definition of* $U$]—yes, the first condition should be
$\exists z$, s.t. $C_\sigma(H, h, z) = h^*$, and [(5)/(9) *definition of* $\Sigma_{const}, \Sigma_{global}, \Sigma_{gvs}, \Sigma_{local}, \Sigma_{lvs}$]—we will revise each of these
definitions by moving the existential quantifier before the universal quantifier.

**R2**: Suggested improvements and regularity/non-regularity properties of the general teaching parameter

**A**: We will add a detailed discussion about these interesting questions and properties mentioned by the reviewer. Below,
we share a few thoughts:

• First, after reading the review, we explored the question of finding upper/lower bounds on the $\Sigma$-TD parameter. We
are able to show that for certain hypothesis classes, $\Sigma$-TD is lower bounded by a function of VCD. In particular, for

the power set class of size $d$ (which has $\mathsf{VCD} = d$), $\Sigma$-TD is lower bounded by $\Omega\left(\frac{d}{\log d}\right)$. We will further study
whether this bound is tight.

• Regarding the additive/sub-additive property, we will continue to study this property and add a detailed discussion in
the revised paper.

• Regarding extension to infinite VC classes, our current results (Lemma 4) is not directly applicable; however, we
consider a generalization to the infinite VC classes as a very interesting direction for future work.

**R4**: *Notions of collusion-freeness in sequential models*

**A**: Collusion-freeness for the batched setting is well established in the research community. It remains an open question
for the research community to agree on a well-accepted notion of collusion-freeness for the sequential setting. In this
paper, we are introducing a possible notion of collusion-freeness for the sequential setting (Definition 1). As discussed
in Section 6, a stricter condition is the "win-stay lose-shift" model, which is easier to validate without running the
teaching algorithm. In contrast, the condition of Definition 1 is more involved in terms of validation and is a joint
property of the *teacher-learner pair*. We will further add a discussion on this in the updated version of the paper.

**R4**: *Discussion on the "presumably increased complexity of sequential learners"*

**A**: Our model generalizes classical teaching models [ZLHZ11, GRSZ17, CSMA+18], and inherits the complexity
results from all these settings. It is known that the optimal teacher achieving TD amounts to solving a set cover problem
which is NP-hard; moreover, the complexity of the sequential teaching has been discussed in [CSMA+18] as a planning
problem. It remains an open problem to understand the complexity of the general sequential teaching setting as a
sequential optimization problem. We will add a discussion in the updated version of the paper.

[Meta-Review · NeurIPS 2019]

It is a long-standing open question whether a preference-based teaching strategy can achieve teaching complexity linear in VC dimension. This work shows that, if we add a sequential aspect to the setting, where the learner's preference function can change after each example from the teacher (in a certain constrained way that avoids extreme "coding tricks"), then the teaching complexity is bounded by the VC dimension. This represents a major advance in our understanding of the complexity of machine teaching. The intermediate lemmas may also be of independent interest. The reviewers all agree that this is solid work, definitely worth accepting.